# Mechanism of the electrochemical hydrogenation of graphene

Y.-C. Soong [1,2,11], H. Li[1,11], Y. Fu [1,2,11] ✉, J. Tong [1,2], S. Huang [1,3], X. Zhang[1,2],
E. Griffin [1,2], E. Hoenig[1,2], M. Alhashmi[1,2], Y. Li[4,5], D. Bahamon[4,5], J. Zhong[6],
A. Summerfield [2], R. N. Costa Filho[7], C. Sevik[8], R. Gorbachev [1,2], E. C. Neyts [9],
L. F. Vega [5,6], F. M. Peeters [7,8,10] & M. Lozada-Hidalgo [1,2] ✉

The electrochemical hydrogenation of graphene induces a robust and reversible conductor-insulator transition, of strong interest in logic-and-memory applications. However, its mechanism remains unknown. Here we show that it proceeds as a reduction reaction in which proton adsorption competes with the formation of $H_2$ molecules via an Eley-Rideal process. Graphene's electrochemical hydrogenation is up to $10^6$ times faster than alternative hydrogenation methods and is fully reversible via the oxidative desorption of protons. We demonstrate that the proton reduction rate in defect-free graphene can be enhanced by an order of magnitude by the introduction of nanoscale corrugations in its lattice, and that the substitution of protons for deuterons results both in lower potentials for the hydrogenation process and in a more stable compound. Our results pave the way to investigating the chemisorption of ions in 2D materials at high electric fields, opening a new avenue to control these materials' electronic properties.

The hydrogenation of graphene was predicted to turn the material into an insulator[1–3], but this has been difficult to achieve experimentally. Graphene was first hydrogenated using accelerated hydrogen atoms or protons in plasmas and heated hydrogen gas sources[4–9]. This yielded samples with notably higher electrical resistivity than pristine graphene, but they were typically not electrically insulating. Normally, only one hydrogenation-dehydrogenation cycle was demonstrated. The mechanism for hydrogenation using accelerated hydrogen atoms has been studied extensively[1,3,10–13]. Theory and scattering experiments demonstrated that a proton approaching a carbon atom in graphene faces an energy barrier of about ~0.3 eV, which if overcome leads to its adsorption in a ~1 eV deep energy well[11,12] (Fig. 1d and Supplementary

Fig. 1). The adsorption probability of an incoming proton then depends on its momentum, which determines whether it overcomes the adsorption barrier, or is reflected by the lattice[1,3,10–13]. On the other hand, recent experiments have demonstrated the hydrogenation of graphene via the electrochemical adsorption of protons[14,15]. This method triggers a robust conductor–insulator transition in graphene that enables field-effect transistors with orders-of-magnitude on-off current ratios. Moreover, hydrogenation via this method is fully reversible and enables precise and robust control of the conductor–insulator transition, such that one million current switching cycles have been demonstrated[14]. These stark phenomenological differences suggest that the mechanisms behind the two methods

[1]Department of Physics and Astronomy, The University of Manchester, Manchester, UK. [2]National Graphene Institute, The University of Manchester, Manchester, UK. [3]University of Bath, Bath, UK. [4]Research and Innovation Center on CO2 and Hydrogen (RICH Center) and Chemical and Petroleum Engineering Department, Khalifa University of Science and Technology, Abu Dhabi, United Arab Emirates. [5]Research and Innovation Center for Graphene and 2D Materials (RIC2D), Khalifa University of Science and Technology, Abu Dhabi, United Arab Emirates. [6]Department of Chemistry, The University of Manchester, Manchester, UK. [7]Departamento de Física, Universidade Federal do Ceará, Fortaleza, Ceará, Brazil. [8]Departement Fysica, Universiteit Antwerpen, Antwerp, Belgium. [9]Department of Chemistry, University of Antwerp, Antwerp, Belgium. [10]Nanjing University of Information Science and Technology, Nanjing, China. [11]These authors contributed equally: Y.-C. Soong, H. Li, Y. Fu. ✉e-mail: yangming_fu@outlook.com; marcelo.lozadahidalgo@manchester.ac.uk

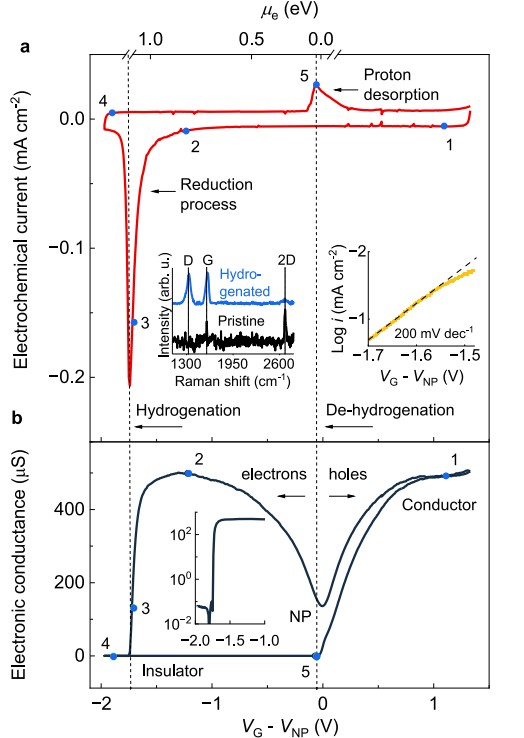

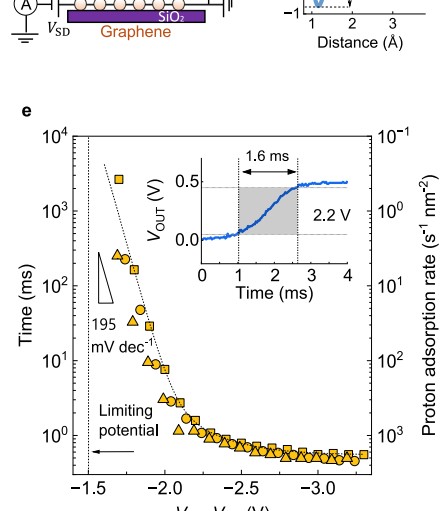

**Fig. 1 | Electronic and electrochemical characterisation of the hydrogenation of graphene. a** Current density vs voltage ($V_G$) from a typical device. Numbered blue dots mark specific points during the voltage sweep. Dashed vertical lines, position of the conductor–insulator transition (hydrogenation) and its reversal (dehydrogenation). Top $x$-axis, Fermi energy vs neutrality point (NP). Left inset, Raman spectra of pristine and hydrogenated samples. The background signal from the electrolyte was subtracted; the latter spectrum was divided by 5 for clarity. Right inset, Tafel plot of the reduction process; voltage sweep rate, 2 mV s⁻¹. Dashed line, Tafel slope. **b** In-plane electronic conductance as a function of $V_G$ obtained simultaneously with the data in (**a**). Graphene is doped with electrons (holes) for negative (positive) potentials vs neutrality point (NP). Inset, zoom-in on the

hydrogenation transition. Numbered blue dots correspond to the same points along the $V_G$ sweep as in (**a**). See Supplementary Table 1 for statistics from different samples. **c** Experimental setup. **d** DFT calculation of potential energy vs distance for a proton approaching a carbon atom in graphene. **e** Time necessary to achieve the insulating transition as a function of $V_G$ (left $y$-axis). Different symbols, data from three different devices. Right $y$-axis, corresponding proton adsorption rate. Triangle marks the Tafel slope of the process. Vertical dashed line marks the minimum potential required to observe the hydrogenation transition. Inset, typical data from experiment showing the rise time of $V_{SD}$ (10–90%) from low to high resistance state (marked with shaded area). $V_G = 2.2$ V.

could be different. In this work, we investigate the mechanism for the electrochemical hydrogenation of graphene. We reveal that it proceeds via a proton reduction mechanism that is up to $10^6$ times faster than previous hydrogenation methods and that it can be tuned by isotope substitution and nanoscale lattice corrugations.

## Results

### Experimental devices

The devices for this work consisted of mechanically exfoliated graphene supported on SiO₂ or hexagonal boron nitride substrates. The devices were then coated with a non-aqueous proton conducting electrolyte with a wide (>4 V) electrochemical stability window (bis(-trifluoromethane)sulfonimide, HTFSI, dissolved in polyethylene glycol, PEG[14,15]) and connected into the electrical circuit as shown schematically in Fig. 1c and Supplementary Fig. 2. For measurements, a voltage was applied to graphene with a PdH$_x$ electrode, which drives an electrochemical current on the surface of graphene. The potential of graphene, $V_G$, is monitored vs a PdH$_x$ pseudo-reference electrode. An additional in-plane source-drain voltage, $V_{SD}$, was applied to graphene to measure its in-plane electronic conductivity as a function of $V_G$ and electronic and electrochemical currents were measured simultaneously. The electronic response of graphene during the hydrogenation process is well established and is used here as a reference for the electrochemical signal of the process. This allows referencing the electrochemical response of the devices directly to the Fermi energy of electrons in graphene (their electrochemical potential).

### Electronic response

Figure 1b shows that the in-plane electronic conductance as a function of $V_G$ displays a local minimum, the so-called charge neutrality point (NP), which separates the hole- and electron-doped branches of the response. For negative potentials vs NP, where graphene is doped with electrons, we observe a conductor–insulator transition when the Fermi energy of electrons is about $\mu_e \approx 1.1$ eV vs NP. This transition is accompanied by a sharp $D$ band in graphene's Raman spectrum, indicating the presence of adatoms in graphene (Fig. 1a, left inset). Previous work has established that this conductor–insulator transition is due to the electrochemical hydrogenation of graphene[14,15], which results in a density of adsorbed protons of ~$10^{14}$ cm⁻². This insulating state could be reversed by applying $V_G \approx 0$ V vs NP, such that graphene recovered its electronic conductivity, and the $D$ band disappeared completely. This allows estimating the Fermi energy for the dehydrogenation transition as $\mu_e \approx 0$ eV vs NP (Estimation of Fermi energy in "Methods" section).

### Electrochemical response

Figure 1a shows that the electrochemical response was dominated by two peaks whose position matched the potentials for the hydrogenation and dehydrogenation processes in the electronic response. The peak at positive potentials was the smaller of the two and could be readily attributed to the oxidative desorption of protons from graphene. However, the peak at negative potentials had an area ~5 times larger than the oxidation peak and consisted of an exponential decay truncated at the insulating transition. Measurements of this reduction

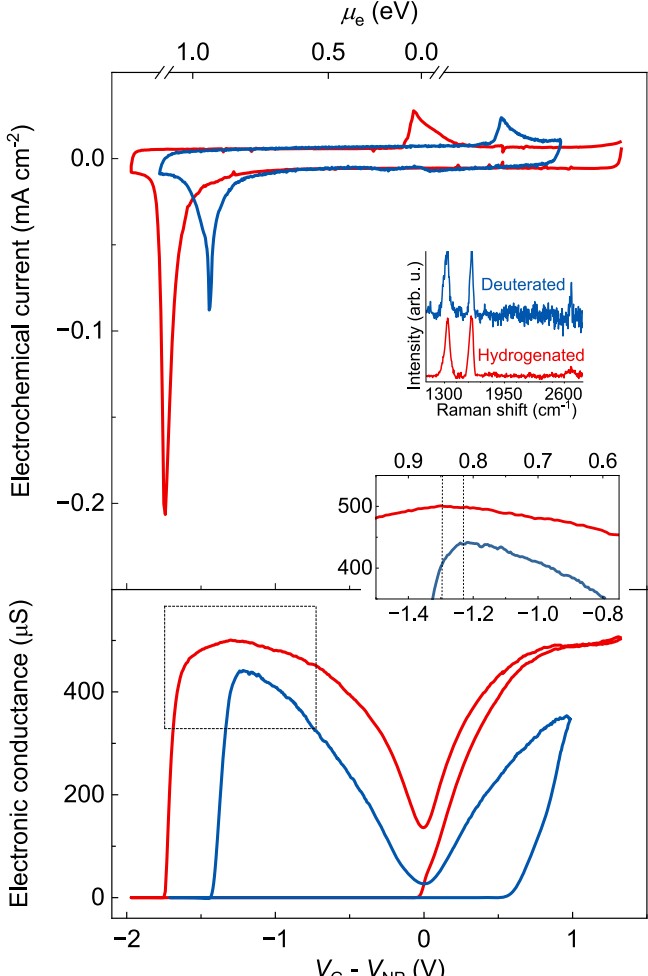

**Fig. 2 | Isotope effect in the electrochemical hydrogenation of graphene.** Top panel, electrochemical response of devices. Top $x$-axis, Fermi energy of electrons in graphene vs NP. Top inset, Raman spectra of hydrogenated (red) and deuterated graphene (blue). Bottom panel, in-plane electronic response of devices measured with proton (red data) and deuteron (blue data) conducting electrolytes. Data obtained simultaneously with data in the top panel. Dashed lines mark a 40 meV difference in Fermi energy. Bottom inset, zoom in from the boxed area in the main panel.

current at slow scan rates revealed that it accelerated exponentially with applied bias as 200 mV dec$^{-1}$ (Fig. 1a right inset, Tafel slope analysis in "Methods" section), a value similar to that observed in graphite electrodes during the hydrogen evolution reaction[16]. These findings therefore show that the reduction peak cannot be attributed only to the hydrogenation of graphene, since in such a case, both oxidation and reduction peaks would be similar. We must conclude that the reduction current is the result of a more complex electrochemical process, which can be attributed to H$_2$ evolution via either an Eley–Rideal or Langmuir–Hinshelwood process[17,18]. In the Eley–Rideal mechanism, an adsorbed hydrogen atom spontaneously combines with an incoming proton and an electron to form H$_2$ (Supplementary Fig. 3). In the Langmuir–Hinshelwood process, two adsorbed hydrogen atoms combine to form H$_2$, but this involves a notable energy barrier (Supplementary Fig. 4). On this basis, we attribute the competing process to an Eley–Rideal mechanism, which proceeds spontaneously. Note that the reduction process would have normally continued beyond the hydrogenation potential, similar to the hydrogen evolution reaction. However, hydrogenation turns graphene into an insulator, yielding the truncated peak observed.

To gain more insights into the reduction reaction, which includes both proton adsorption and the competing process, we measured the rate of proton adsorption directly, isolating it from all competing processes. To that end, we fixed $V_G$ and measured the in-plane electronic response of graphene as a function of time with sub-millisecond resolution, as shown in Fig. 1e and Supplementary Fig. 5. Since the hydrogenation process stops[14,15] when the proton concentration on graphene reaches ~10$^{14}$ cm$^{-2}$, the time $\tau$ at which this transition is reached allows estimating the rate of proton adsorption directly as $r \approx 10^{14}$ cm$^{-2}$ $\tau^{-1}$. Figure 1e shows that for potentials around −1.5 V vs NP, the rate was $r$~1 s$^{-1}$ nm$^{-2}$. For reference, we recall that accelerated hydrogen atom experiments report[9,19] an adsorption rate $r$~10$^{-3}$ s$^{-1}$ nm$^{-2}$, whereas ultra-sensitive gas microcontainer experiments[20,21] report $r$~10$^{-8}$ s$^{-1}$ nm$^{-2}$. Hence, even at low potentials, the electrochemical hydrogenation proceeds orders of magnitude faster than other methods. Notwithstanding, the rate in our experiments accelerated exponentially with bias, such that for −2.5 V vs NP, it reached $r$~10$^{3}$ s$^{-1}$ nm$^{-2}$, which is 10$^6$ times faster than in experiments with accelerated hydrogen atoms. From these data, we also find that $r$ accelerates exponentially with the applied bias as ≈195 mV dec$^{-1}$, which is the same acceleration (Tafel slope) found for the whole reduction process (Fig. 1a). This suggests that proton adsorption is the limiting step of the reduction process. At even larger potentials, $\tau$ saturated at a value of about 10 times the resistor-capacitor (RC) constant of the circuit, which suggests that charging the electrochemical double layer becomes the limiting factor for the process at high bias (Device fabrication in "Methods" section).

## Isotope effect

To confirm that proton adsorption is the limiting step of the reduction process, we measured our devices using electrolytes in which protons were substituted for deuterons, nuclei of hydrogen's heavier isotope deuterium. The deuterated electrolyte was synthesised, producing an electrolyte with the same conductivity as in its proton form[22] (Supplementary Fig. 6, 7). Figure 2, bottom panel, shows that deuterated samples also displayed a conductor–insulator transition, and that their Raman spectra were the same as for hydrogenated graphene (Fig. 2, top inset), demonstrating a deuterium concentration of ~10$^{14}$ cm$^{-2}$ on graphene. However, graphene's deuteration took place at less negative potentials compared to its hydrogenation. This finding was confirmed with in situ Raman measurements (Supplementary Fig. 8) and is consistent with previous experiments of graphene's deuteration using accelerated atoms, which found that the deuteration of graphene proceeded faster than its hydrogenation[19]. To quantify this isotope effect, we studied it in the limit of low adsorbed proton/deuteron density, which can be accessed in the potential range at which graphene's electronic conductivity started to decrease (Fig. 2, bottom inset). This revealed that the onset of the deuteration transition occurred for a Fermi energy several tens of meV lower than with the proton electrolyte.

Figure 2, top panel, shows that both electrochemical peaks shifted to the positions of the deuteration and de-deuteration transitions. We also found that their relative sizes changed. The oxidation peak had roughly the same area for samples measured with proton and deuteron electrolytes, but the reduction peak had half the area for samples measured with deuteron electrolyte. Since both types of samples achieve the same concentration of adsorbed protons/deuterons (~10$^{14}$ cm$^{-2}$), the smaller reduction peak demonstrates that the competing (Eley–Rideal) process is suppressed for the deuteron reduction process. This shows that the hydrogenation process is faster when the competing process is suppressed, which confirms that proton/deuteron adsorption is the limiting step in the reduction process. From a theory perspective, the isotope effect can be traced back to differences in the zero-point energy of the carbon-hydrogen and carbon-deuterium bonds, which lead to a zero-point energy ≈45 meV lower for

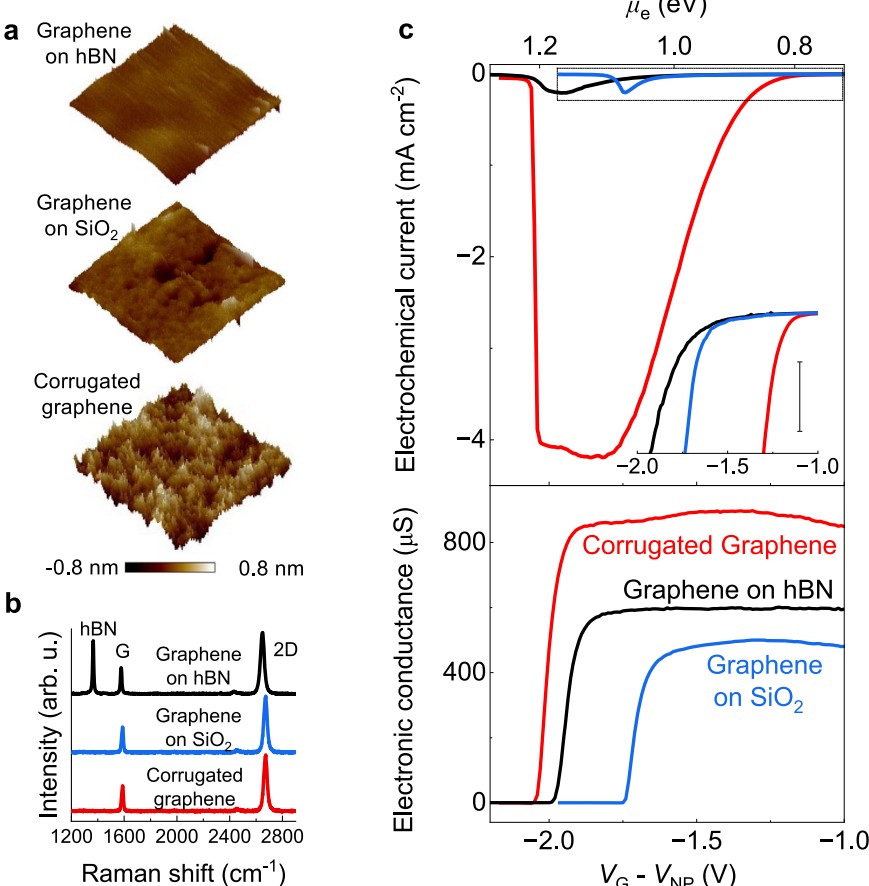

**Fig. 3 | Activation of graphene's basal plane towards proton reduction via nanoscale corrugations. a** AFM topography maps (200 nm × 200 nm) of the three types of devices. **b** Raman spectra of devices in (**a**) (not in contact with the electrolyte). **c** Top panel, electrochemical current of devices. Inset, zoom in of reaction onset taken from the boxed square in the main panel. Scale bar, 0.1 mA cm⁻². Bottom panel, electronic conductance of devices on the electron-doped branch of the response.

deuterons than for protons[19,23]. This makes deuterons harder to remove from graphene, in agreement with our observations. Hence, taken together, our experimental findings demonstrate that hydrogenation takes place as part of a reduction process that involves proton adsorption as the limiting step and that the adsorption process competes with the formation of $H_2$ via an Eley–Rideal process. From the perspective of applications, the isotope substitution experiments show that the deuteration of graphene has the advantage of yielding an insulating transition at lower potentials than the hydrogenation process and results in an even more stable insulating state.

## Electrochemical potentials

Our results establish the mechanism of the hydrogenation process and its relation to the electronic response of the devices. However, the electrochemical potential for the hydrogenation found experimentally, $\mu_e \approx 1$ eV, would initially appear to differ from the DFT calculations, which report a small energy barrier (-0.3 eV) for this process (Fig. 1d). To reconcile these observations, we note that the DFT calculations assume protons to be in vacuum, whereas protons in the experiments at zero bias are in equilibrium with their adsorption and desorption in the $PdH_x$ electrode[24]. Moreover, the experiments measure changes in electrochemical potential during the process, rather than in energy. To account for these differences, we calculated the Gibbs energy of a proton during the hydrogenation process, in the neutrality point (NP) scale, using the formula[25–27]: $\Delta G = U_H - U_{ref} + \Delta ZPE - T\Delta S - \Delta NP$. In the formula, $U_H$ is the proton adsorption energy in vacuum calculated by DFT; $U_{ref}$ is the energy of protons in the $PdH_x$ electrode in the standard hydrogen scale[25,28], ΔZPE and $\Delta S_H$ are the differences in zero-point energy and entropy between the proton in the adsorbed state and the reference, respectively; and ΔNP is the position of graphene's neutrality point in the standard hydrogen electrode scale. When all these contributions are included, the DFT calculation translates into an electrochemical potential of ≈1.4 eV vs NP for the hydrogenation and ≈0.1 eV vs NP for the dehydrogenation (Analytical theory in Methods). This analysis, therefore, reconciles the DFT calculations with the Fermi energy extracted experimentally within an error of only a couple of hundred meV.

## Control of processes via graphene's morphology

Previous works suggest that changing graphene's morphology at the nanoscale could alter its electrochemical properties, even in the absence of defects such as vacancies, adatoms or substitutions[21,29–31], but this has not been investigated for the electrochemical hydrogenation of graphene. To investigate this possibility, we fabricated two new types of devices. In the first one, graphene was supported on hexagonal boron nitride, rather than on silicon oxide, which yields atomically flat graphene[21,32] (Fig. 3a). In the second type, we increased the roughness of graphene at the nanoscale by subjecting our standard graphene devices to thermal cycling between extreme temperatures, resulting in corrugated graphene. Typically, we heated the devices to 400 °C, then rapidly cooled them to liquid nitrogen temperatures (−196 °C) within seconds and repeated this treatment 3–4 times. Graphene's negative thermal expansion coefficient[33] results in a strong mismatch in thermal expansion with the substrate. This increases

graphene's mean square roughness by a factor of ~2 (Fig. 3a, Supplementary Fig. 9 and Corrugated graphene in "Methods" section), and probably even more than the AFM measurements show, since tip convolution effects in scanning probe techniques are known to underestimate the aspect ratio of nanoscale features[34]. Crucially, this thermal treatment did not introduce any cracks or microscopic defects, as evidenced both in the AFM images and from the lack of a $D$ band in the Raman spectra of the samples (Fig. 3b).

Figure 3c shows that the electrochemical current in the different samples displayed strong differences. Corrugated graphene exhibited the lowest onset potential for the reduction process, followed by graphene-on-SiO$_2$, and finally graphene-on-hBN (inset, Fig. 3c). As a result, the reduction current in corrugated graphene reached values approximately 20 times higher than those in graphene-on-SiO$_2$ devices. In situ Raman measurements confirmed that the hydrogenation of the corrugated samples was also accompanied by a sharp $D$ band (Supplementary Fig. 10). Our DFT calculations indicate that this occurs because corrugations effectively eliminate the energy barrier for proton adsorption in convex regions of the flake, in agreement with previous work[13,21,29,30] (Supplementary Fig. 1). This facilitates proton adsorption and thus results in a larger reduction current via an Eley–Rideal process. In contrast, the reduction current was suppressed for graphene-on-hBN devices, which arises because its atomic flatness increases the proton adsorption barrier. Unexpectedly, we found that despite the lower onset potential for proton adsorption in corrugated graphene, the insulation transition occurred at similar potentials across all samples. This shows that achieving a proton coverage of ~10$^{14}$ cm$^{-2}$ required comparable potentials in all cases. We attribute this to regions within the corrugated graphene that resist hydrogenation, preventing the entire sample from reaching full proton coverage. Our DFT calculations support this interpretation, showing that while proton adsorption barriers are lower in convex areas, they remain the same as in flat graphene in concave regions. These concave areas likely maintain electronic conductivity, and as a result, the conductor–insulator transition of the entire flake is governed by proton adsorption in these regions.

## Discussion

Our work elucidated the mechanism behind the electrochemical hydrogenation of graphene, revealing the role of competing processes, isotope effects and material morphology. We demonstrated that graphene's hydrogenation competes with the spontaneous formation of H$_2$ molecules; that it proceeds 10$^6$ times faster than hydrogenation with accelerated hydrogen atoms; and that its reversal (dehydrogenation) proceeds via the oxidative desorption of protons. These results were explained using first-principles theory. Our work provides a framework to control the electrochemical chemisorption of protons in graphene at high electric fields and could be expanded to ions such as fluorine[35] or oxygen[36]. This paves the way for the control of graphene's electronic properties in novel chemistry-based computing devices[15,37,38].

## Methods

### Device fabrication
Source and drain electrodes (Au/Cr) were patterned using photolithography and electron beam evaporation on Si/SiO$_2$ substrates. Mechanically exfoliated monolayer graphene crystals with a rectangular shape were transferred onto the substrates to form a conducting channel between the source and drain electrodes. An SU-8 epoxy washer with a 15-μm diameter hole covered the source and drain electrodes, leaving a mid-section of the graphene flake exposed (Supplementary Fig. 2). This allows the whole mid-section of the flake to be hydrogenated, which is necessary to observe the insulating transition. The electrolyte used was 0.18 M bis(trifluoromethane)sulfonimide (HTFSI or DTFSI) dissolved in poly(ethylene glycol) (average

molecular weight Mn of 600), which was drop-cast on the device inside a glovebox containing an inert gas atmosphere. We characterised the conductivity of both HTFSI and DTFSI electrolyte by measuring the conductance of devices consisting of SiN$_x$ substrates with a circular hole of different diameters. This allows us to extract the electrolyte conductivity, $\kappa$, from the measured conductance, $G$, using the equation[39]: $G = 4\pi\kappa r$. This reveals $\kappa \approx 1.062 \times 10^{-4}$ S cm$^{-1}$ for HTFSI, consistent with other studies[40–42] and a≈35% lower conductivity in the DTFSI electrolyte. Together with the capacitance of the electrolyte we characterised in previous work, $C \approx 20$ μF cm$^{-2}$, we estimate the RC time constant of our microelectrode devices as ≈0.6 ms. Palladium hydride or palladium deuteride foils (~0.5 cm$^2$) fabricated following the recipe in ref. [43] were used both as gate and pseudo-reference electrodes. The reaction taking place in the gate and reference electrode is proton adsorption into the metal xH$^+$ + xe$^-$ + Pd → PdHx[24,44,45]. The device was placed inside a chamber filled with 100% hydrogen gas for electrical measurements.

### Electrical and electrochemical measurements
For electrochemical measurements, we used a Keithley 2636B source meter and swept the potential at a rate of 10 mV s$^{-1}$ using programmes that allow controlling the potential of graphene vs the pseudo-reference electrode, $V_G$. We found that $V_G$ is effectively the same as the applied potential, within a scatter of <4 mV, similar to a previous work[15,46]. Reference experiments with an Ivium pocketSTAT2 potentiostat gave the same results. A second Keithley source meter was used both to apply source-drain bias ($V_{SD}$) and to measure the electronic current. The electrochemical response is normalised by the active area of the devices.

In the manuscript, we have adopted the convention that negative (positive) potentials vs NP dope graphene with electrons (holes). In the setup, protons flow from the gate electrode to graphene during the reduction process (starting at about −1.5 V vs NP), yielding a negative current. Conversely, protons flow from graphene to the gate electrode during the oxidation process (at about 0 vs NP) and yield a positive current.

For measurements of the time dependence of the conductor–insulator transition, graphene was connected to the voltage divider circuit shown in Supplementary Fig. 5, similar to a previous work[14]. Graphene was connected to a 1 MΩ series resistor, and the total voltage was set to 1 V. The gate electrode was connected to an alternating voltage source using a waveform generator that oscillated between a hydrogenating potential, typically set between −1.5 V and −3.3 V vs NP, and a dehydrogenating potential of +1.2 V vs NP. An oscilloscope or a Keithley 2182A Nanovoltmeter was used to record the output voltage across the graphene channel. The conducting and insulting phases are evident from the low and high plateaus of the output voltage. The time constant was defined as the time it took the response to rise from 10% to 90% of these plateaus.

### Tafel slope analysis
The acceleration of an electrochemical process with applied bias is characterised by its Tafel slope. This parameter is defined as[39] $b = \ln(10) \, kT/\alpha e$, with $kT$ the thermal energy, $e$ the elementary charge constant and $\alpha \in [0, 1]$ a coefficient that models the symmetry of the energy barrier of the reaction. From the measured $b \approx 200$ mV dec$^{-1}$, we deduce $\alpha \approx 0.3$. This coefficient implies[39] that the proton reduction reaction is harder than reversing it via the oxidative removal of protons from graphene, consistent with our measurements and energy analysis of these processes.

### Raman spectroscopy
The Raman spectra of devices fabricated on glass substrates were measured as a function of applied bias using a WITec Alpha300R Raman spectrometer equipped with a laser operating at 514.5 nm. For

each bias, the measurement was performed with a 50× objective, 600 l mm⁻¹ grating, accumulating time of 75 s over 4 cycles, and the laser power was 0.5 mW. The raw spectra include an electrolyte background alongside the graphene signal. Since this background is independent of applied voltage, it is removed using the data processing software's (Origin) background subtraction tool. Raman spectra of devices not in contact with electrolyte (Fig. 3) were performed on a Renishaw inVia spectrometer equipped with a laser operating at 532 nm using a 100× objective, 1800 l mm⁻¹ grating, accumulating time of 15 s over 3 cycles with the laser power of 1 mW.

The data show that the hydrogenation transition was accompanied by the appearance of a strong $D$ band, a sharp increase of the G peak intensity and the smearing of the 2D peak. This demonstrates high disorder in graphene[47]. The density of adsorbed hydrogen atoms in hydrogenated graphene was estimated from the intensity ratio of the $D$ and $G$ bands[47], $I_D/I_G$, and the integrated-area ratio of $A_D/A_G$. From the found $I_D/I_G \approx 1$, $A_D/A_G \approx 2.03$ and the fact that the $2D$ band is smeared, a density of H atoms of $\approx 1 \times 10^{14}$ cm⁻², was estimated, in agreement with previous reports[14,15]. Note that $I_D/I_G \approx 1$, $A_D/A_G \approx 2.03$ can translate both into a distance between H atoms of $L_D \approx 1$ nm or $L_D \approx 10$–15 nm[47,48]. This is a consequence of the bell-shaped $I_D/I_G$ vs $L_D$ dependence[47,48]. To decide which one applies, it is necessary to look for additional signatures of disorder in the spectra. In our spectra, the $2D$ band is smeared, a signature of highly disordered systems, and thus consistent with $L_D \approx 1$ nm, or a defect concentration of $\approx 1 \times 10^{14}$ cm⁻². The Raman spectra of pristine graphene were recovered by dehydrogenating the sample, as demonstrated previously[14,15].

Our measurements with deuterium electrolyte resulted in a similar response (Supplementary Fig. 8), except the potential at which the $D$ band appeared was lower, in agreement with the transport experiments.

## Deuterated electrolyte synthesis

Lithium bis(trifluoromethane)sulfonimide (LiTFSI, 99%) and Dideuterosulfuric acid (D₂SO₄, 96–98 wt. % in D₂O, 99.5 atom% D) were purchased from Sigma–Aldrich and used as received without further purification. All manipulations of air- and moisture-sensitive compounds were carried out under an atmosphere of dry nitrogen using standard Schlenk techniques.

Deuterated bis(trifluoromethanesulfonyl)imide (DTFSI) was synthesised by dissolving 5 g (26.75 mmol) LiTFSI in 5 mL D₂SO₄ and heating the mixture to 75 °C in a sublimator under a nitrogen atmosphere. The formed product was sublimed and collected under reduced pressure over a curved glass tube in a liquid nitrogen bath. Note that the melting and decomposition of LiTFSI proceed at 234 °C and 340–415 °C, respectively[49]; whereas the boiling point of D₂SO₄ is 290 °C; and the melting and boiling points of Lithium sulfate are 859 °C and 1,377 °C, respectively. Hence, only the product DTFSI can be sublimed in this process.

To characterise the electrolyte, NMR spectra of the product (DTFSI) were obtained on a Bruker Avance III Prodigy instrument (500 MHz for ¹⁹F NMR, 151 MHz for ¹³C NMR and 400 MHz for ⁷Li NMR). Samples were prepared by dissolving the solid powder (-10 mg) in DMSO-d₆ (1 mL). Chemical shifts (δ) were described in parts per million (ppm) from high to low frequency and referenced to the residual solvent resonance. The residual signal from the chemical shift of DMSO-d₆ at 39.52 ppm was used as an internal reference for ¹³C NMR[50].

Supplementary Fig. 7 shows that the ¹⁹F NMR spectrum of DTFSI displayed one set of singlet characteristic fluorine peak, demonstrating only one fluorine environment; the ¹³C NMR spectrum of DTFSI had only one quartet signal, demonstrating only one carbon environment. The ¹⁹F NMR and ¹³C NMR data illustrate that there was only one set of (CF₃SO₂)₂N-anions in the DTFSI product. To rule out the existence of Li in the DTFSI, we also measured the ⁷Li NMR spectra of LiTFSI and DTFSI. A strong singlet lithium peak at a chemical shift of −1.05 ppm was observed in the ⁷Li NMR spectrum for LiTFSI, whereas no peaks were observed for the DTFSI, demonstrating the absence of Li in the DTFSI sample, further confirming the purity of the product.

## Proton transport in the HTFSI/PdHx system

Protons in our devices originate from trifluoromethanesulfonimide (HTFSI), a superacid (pKₐ ≈ −10), and are present either as free H⁺ or as PEG-coordinated complexes[51,52]. Proton conduction occurs primarily via a Grotthuss-type mechanism, enabled by transient hydrogen bonding with PEG oxygen groups[51,52]. This is consistent with our conductivity measurements in Supplementary Fig. 6, which revealed a clear isotope effect. At the counter electrode, protons interact with palladium, forming Pd hydride (PdHx). The Pd−H system exhibits two phases: a hydrogen-poor α-phase and a hydrogen-rich β-phase, whose distribution depends on hydrogen concentration, temperature, and pressure[44,45]. In acidic aqueous media under H₂ atmosphere, hydrogen-saturated Pd electrodes equilibrate at ≈0 V vs SHE through the Volmer step (H⁺ + e⁻ + * → H*), whereas pristine Pd shows ~90 mV vs SHE, which very gradually decreases as hydrogen is electrochemically absorbed. Electrodes with intermediate hydrogen loading exhibit potentials between 0 and 90 mV. In our devices, the PdHₓ displays a zero current potential of ≈61 mV vs SHE.

These redox equilibria persist in non-aqueous electrolytes, albeit with slower kinetics due to reduced proton mobility and altered solvation environments[53]. The potentials for proton adsorption and desorption can also shift by approximately 100 mV relative to their values in aqueous electrolytes[53]. Nonetheless, the underlying mechanism and equilibrium positions remain comparable to those in the aqueous case. In our devices, the electronic response of graphene provides direct access to the absolute potential scale, effectively circumventing these small uncertainties.

## Analytical theory

To calculate the electrochemical potential of the protons during the hydrogenation process, we need numerical values for the different entries in the formula[25]

$$\Delta G = U_H - U_{ref} + \Delta ZPE - T\Delta S - \Delta_{NP} \tag{1}$$

The first term $U_H \approx$ -0.7 eV to −1.08 eV is the energy of a proton adsorbed on graphene vs vacuum, calculated by DFT[12]. The second term, $U_{ref}$, is the energy of a proton adsorbed on the reference electrode vs SHE, given by $U_{ref} = \frac{1}{2} U_{H2} - U_{PdH}$, with $-U_{H2} = 4.47$ eV the binding energy of a hydrogen molecule[25] and $-U_{PdH} = 60$ meV the energy of electrons in the PdHₓ pseudo-reference electrode during the adsorption process in the SHE scale. The third term, ΔZPE, is the difference in zero-point energy between the proton adsorbed on graphene, $ZPE_{C-H}$, and the reference, $ZPE_{H2}$. Hence[25] $\Delta ZPE = ZPE_{C-H} - \frac{1}{2} ZPE_{H2} \approx 0.17$ eV $- \frac{1}{2} \times 0.27$ eV $= 0.035$ eV. The fourth term, $T\Delta S$, is dominated by the contribution from the gas phase, so it is given by[25] $T\Delta S \approx \frac{1}{2} T\Delta S_{H2} \approx -0.205$ eV. The fifth term, ΔNP, is the position of the neutrality point in the SHE. To calculate it, we note that 0 vs SHE corresponds to $\varepsilon_0^{abs} \approx 4.44$ eV in the so-called absolute scale[39], in which the energy of electrons is referred against their energy in vacuum. The energy necessary to remove an electron from the neutrality point in graphene into the vacuum is[54] $F^G \approx 4.5$ eV. Hence, $\Delta_{NP} = \varepsilon_0^{abs} - F^G \approx -50$ meV. This yields $\Delta G \approx 1.4$ eV vs NP, which is in reasonable agreement with the experiment. The dehydrogenation potential is estimated with this formula from the energy necessary for the proton to escape the adsorption well (-1.3 eV).

## Isotope substitution experiments

Using formula (1), we calculated $\Delta G_H - \Delta G_D = (U_H - U_D) - (U_{ref}^H - U_{ref}^D) + (\Delta ZPE^H - \Delta ZPE^D) - (T\Delta S^H - T\Delta S^D)$, where sub- and super-indexes denote protons (H) and deuterons (D). This expression can be

simplified by noting that $\Delta ZPE^H - \Delta ZPE^D = (ZPE_{C-H} - ZPE_{C-D}) - (ZPE_{H2} - ZPE_{D2})$ and that $(ZPE_{H2} - ZPE_{D2}) = -(U_{ref}^H - U_{ref}^D)$. Moreover, $U_H = U_D$, because this is the energy calculated by DFT. Neglecting changes in the entropy of hydrogen and deuterium molecules, we obtain the formula:

$$\Delta G_H - \Delta G_D \approx ZPE_{C-H} - ZPE_{C-D} \qquad (2)$$

To obtain the ZPE for the carbon-proton bond, we note that the vibrational energy is given by $h\nu$, with $h$ the Planck constant and $\nu$ the vibrational frequency of the bond. From the known $\nu = 2738$ cm$^{-1}$ of this bond, we obtain its ZPE as $h\nu/2 = 0.17$ eV. For deuterons, we obtain $h\nu/2 = 0.124$ eV. This then yields an isotope effect of $\Delta G_H - \Delta G_D \approx 45$ meV.

To measure the isotope effect experimentally, it is necessary to measure the proton/deuteron adsorption in the linear regime of the process[55]. This is difficult for the hydrogenation process because this process only takes place at high applied biases and is only measurable when a relatively large number of protons are adsorbed on the surface of graphene. To address this challenge, we extracted the Fermi energy for the potential at which the electronic conductivity of the samples starts to decrease. Data from 12 different samples show that this energy was ≈0.87 eV vs NP for protons and ≈0.82 eV vs NP for deuterons. However, given the large experimental uncertainty in extracting this parameter, we can only estimate that the onset potential of the deuteration transition is several tens of meV lower for deuterons.

### Estimation of Fermi energy and carrier mobility

The (gate) potential applied to graphene is related to the charge carrier density $n$ as[56,57]: $V_G - V_{NP} = neC^{-1} + \hbar\nu_F\,e^{-1}(\pi n)^{1/2}$, where $C$ is the capacitance of the electrolyte, $\nu_F \approx 1 \times 10^6$ m s$^{-1}$ the Fermi velocity in graphene, and $\hbar$ the reduced Planck constant. Since $\mu_e = \hbar\nu_F\sqrt{\pi n}$, this yields the relation between potential and Fermi energy: $V_G - V_{NP} = eC^{-1}\pi^{-1}\,\mu_e^2\,(\hbar\nu_F)^{-2} + e^{-1}\mu_e$. In previous work[15], we have estimated the capacitance of the electrolyte used in this work experimentally as $C \approx 20\ \mu F\ cm^{-2}$, which enables a direct numerical estimate of $\mu_e$. Note that this description is valid only if the system is conducting and outside a bandgap[57]. For this reason, we cannot specify $\mu_e$ after the hydrogenation transition, as indicated by breaks in the top axes in Figs. 1–3. In principle, this precludes estimating $\mu_e$ for the dehydrogenation transition. However, from the position of the NP before the hydrogenation transition and given the high reproducibility of the dehydrogenation potential vs the NP, we tentatively estimate the Fermi energy for the dehydrogenation as $\mu_e \approx 0$ eV vs NP.

We also estimated the carrier mobility in our samples using the protocols reported previously. The carrier concentration in graphene, $n$, is:

$$n = \sqrt{n_0^2 + n(V_G^*)^2} \qquad (3)$$

where $n_0$ is the residual charge at the NP and $n(V_G^*)$ are the induced carrier concentration as a function of the applied gate voltage, $V_G$ vs the neutrality point. The relation between gate voltage and induced carriers is:

$$V_G^* = \frac{e}{C} \cdot n\left(V_G^*\right) + \frac{\hbar \cdot \nu_F \cdot \sqrt{\pi n(V_G^*)}}{e} \qquad (4)$$

where $e$ and $\hbar$ are the elementary charge and reduced Planck constant, respectively. $\nu_F$ is the Fermi velocity, of $1 \times 10^6$ m s$^{-1}$ and $C = 20\ \mu F\ cm^{-2}$ is the electrolyte capacitance, as shown in our previous work. The total resistance, $R_{Total}$, is then written as:

$$R_{Total} = R_C + \frac{L_G}{W_G} \cdot \frac{1}{e \cdot \mu \cdot n} \qquad (5)$$

where $R_C$ is the contact resistance and $L_G$ and $W_G$ refer to the gated channel length and width, respectively, and $\mu$ is the mobility to be extracted. The extracted mobilities for graphene-on-SiO$_2$ and corrugated graphene devices are in the range of 2000–6000 cm$^2$/V·s, in good agreement with the literature for similar samples, which display mobility between 1000 and 10,000 cm$^2$/V·s[58]. For Gr-on hBN devices, the mobility values are ~15,000–30,000 cm$^2$/V·s, also in agreement with previous studies[32].

### DFT calculations of proton adsorption

Density functional theory (DFT) calculations were carried out using the plane-wave-based DFT method as implemented in the Vienna ab Initio Simulation Package (VASP)[59]. DFT calculations with a dispersion correction method (DFT-D3) were performed to account for van der Waals interactions, using Grimme's method to correct for long-range interactions[60]. Spin-polarisation was included in the simulations. The generalised gradient approximation with the Perdew-Burke-Ernzerhof functional (GGA-PBE) was used to calculate the total energy[61]. Electron-ion interactions were described using the projector augmented wave (PAW) method[62]. An energy cutoff of 500 eV was set for the plane-wave basis. The convergence criteria were set to $10^{-6}$ eV and 0.01 eV Å$^{-1}$ for the electronic self-consistent iteration and the forces on each atom, respectively. The Brillouin zone was sampled using Monkhorst-Pack mesh k-points with a reciprocal space resolution of $2\pi \times 0.04$ Å$^{-1}$. The computational box used to model graphene had an orthorhombic shape with periodic boundary conditions in all three directions.

The graphene layer consisted of a supercell with $7 \times 4$ unit cells with a total number of 112 carbon atoms. A periodic length of $z = 20$ Å was considered along the z-direction perpendicular to the graphene plane, with the graphene being located at the centre in the z-direction to avoid self-interactions with periodic images. To evaluate the effect of curvature on the hydrogenation processes, graphene structures with different concavities were generated. Corrugations were modelled by fixing the out-of-plane positions of carbon atoms so that the crystal forms a Gaussian profile of height $h$ (Supplementary Fig. 1). The atoms were allowed to relax in-plane.

Supplementary Fig. 1 shows the energy profiles for the proton as a function of the aspect ratio of the corrugated graphene structures. We find that for convex structures, the energy barrier for the incoming proton is lower and the adsorption well is deeper with respect to flat graphene, whereas for concave structures, the energy profiles are effectively unchanged, in agreement with previous calculations[29,30].

### Eley–Rideal process

The potential energy surface (PES) for the formation of H$_2$ on flat graphene via an Eley–Rideal process was calculated with DFT using the same parameters described above. In this process, an incoming proton (H$_t$) approaches a proton adsorbed on graphene (H$_b$). The incoming proton follows the trajectory perpendicular to the basal plane directly on top of the adsorbed proton. In the calculation, the protons were fixed at specific coordinates in the trajectory and then both protons, the carbon atom with the adsorbed proton and its 1$^{st}$ and 2$^{nd}$ neighbours were allowed to relax in the z-direction. We calculated the energy of the whole system (graphene plus both protons) for all these coordinates. 160 points of the PES were calculated as a function of the distances of the two H atoms from the surface, and 2D splines were used to have continuous representations of the functions. Supplementary Fig. 3 shows the PES for the process and the minimum energy path within it. These calculations show that the formation of H$_2$ on graphene via an Eley–Rideal mechanism is a spontaneous process, in agreement with previous work[17,18].

## Langmuir–Hinshelwood process

The formation of $H_2$ on graphene via a Langmuir–Hinshelwood process was calculated using climbing image nudge elastic band (CI-NEB) DFT using the parameters described above. In this process, two protons adsorbed on graphene recombine to form a $H_2$ molecule. These two protons can adsorb in different relative positions, known as para and ortho dimers. Supplementary Fig. 4 shows that for both configurations, the formation of $H_2$ involves a large ~1–2 eV energy barrier, which makes this mechanism energetically unfavourable compared to the Eley–Rideal process.

## Corrugated graphene

Mechanically exfoliated graphene flakes were transferred onto Si/SiO$_2$ substrates with source-drain electrodes, as described above. To corrugate the flakes, the samples were first annealed in a tube furnace in a He atmosphere at 400 °C for a couple of minutes, following the recipe from a previous work[34]. The samples were then pulled out of the furnace and placed into liquid nitrogen for rapid cooling. This annealing process was repeated three times. To characterise the roughness of the samples, we performed ambient atomic force microscopy (AFM) using a Dimension FastScan in peak force mode. For further confirmation, additional before/after thermal cycling images were obtained with a separate AFM system (Oxford Instruments Asylum Research Cypher-S) using Budget Sensors Tap300Al-G probes. Both pre- and post-thermal cycling roughness measurements were acquired in AC mode, and all data was processed using the Gwyddion[63] SPM and NanoScope data analysis package. Supplementary Fig. 9 shows height statistics collected for both types of samples. These revealed that the thermal treatment increased the roughness of the samples from ≈0.1 nm to ≈0.17 nm. The changes in the roughness of the samples can be expected to be larger, since tip convolution effects in scanning probe techniques tend to underestimate the aspect ratio of nanoscale features.

## Data availability

All data supporting the findings of this study are available within the article, the Supplementary Information file or in the database under accession code https://zenodo.org/records/17209896.

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

## Acknowledgements

This work was supported by UKRI (EP/X017745: M.L.-H.), the Directed Research Projects Program of the Research and Innovation Center for Graphene and 2D Materials at Khalifa University (RIC2D-D001: M.L.-H., L.F.V. and D.B.), The Royal Society (URF\R1\201515: M.L.-H.), the U.S. Army DEVCOM ARL Army Research Office (ARO) Energy Sciences Competency, (Electrochemistry or Advanced Energy Materials) Program award # W911NF-25-1-0041 (M.L.-H.). The views and conclusions contained in this document are those of the authors and should not be interpreted as representing the official policies, either expressed or implied, of the U.S. Army or the U.S. Government. Part of this work was supported by the Flemish Science Foundation (FWO-Vl: F.M.P.) and FUNCAP and CNPq (309908/2022-1: RNCF).

## Author contributions

M.L.-H. designed the project and directed it with help from Y.F., H.L. and J.T. Y.-C.S. synthesised and characterised the corrugated graphene with help from A.S. and R.G. J.T., Y.F., H.L. and Y.-C.S. performed electrochemical measurements and analysis with help from X.Z. and S.H. Y.-C.S., Y.F., J.T., E.G. and M.A. fabricated devices with help from S.H., X.Z., and E.H. H.L. and J.Z. synthesised and characterised the deuterated electrolyte. F.M.P., E.C.N. and R.N.C.F. performed analytical theory calculations. D.B., C.S., Y.L. and L.F.V. performed DFT calculations. M.L.-H. and Y.-C.S. wrote the manuscript with input from all the authors.

## Competing interests

The authors declare no competing interests.
