## [Transparent Peer Review file · Nature Communications]

Mechanism of the electrochemical hydrogenation of graphene

Corresponding Author: Professor Marcelo Lozada-Hidalgo

Version 0:

Reviewer comments:

Reviewer #1

(Remarks to the Author)

Referee report for: Mechanism of the electrochemical hydrogenation of graphene

In this manuscript, Y.-C. Soong and co-authors discussed the mechanism of the electrochemical hydrogenation of graphene with the acid electrolyte (HTFSI) dissolved in PEG. They found differences in oxidation and reduction currents during cyclic voltammetry and demonstrated that the hydrogenation proceeds as a reduction reaction in which proton adsorption competes with a process attributable to the formation of H₂ molecules. The work is really interesting and important. I have a few questions that trigger my curiosity.

1. The circuit design involves several electrodes (Figure 1b and Supplementary Figure 2a). Are those all PdHx electrodes? Are there relationships between them, is there an electrochemical current running between them or could one electrode influence the other during the measurements (see attached figure)? It seems to me that there are now two reference electrodes and a bias is applied between the two. Are the gate and the reference electrode biased with respect to each other? In other words, if an electrochemical reaction occurs at the graphene surface and that the graphene is used as a GFET, how to disentangle the two contributions (i.e., electrochemistry and field effect)?
2. Could the authors discuss the mechanism of proton migration in the electrolyte (Grotthuss mechanism, or vehicle mechanism). Supplementary Figure 6 shows that the conductivity of proton (in HTFSI) is smaller than that of deuteron (in DTFSI). However, it is known that the mobility of proton should be greater than that of deuteron because the mass difference.
3. In the manuscript, the graphene located in the SU8 washer opening is getting hydrogenated. Figure S2 suggests that the graphene flake is smaller than the opening in the SU8. Are the graphene samples studied here bigger or smaller than the opening in the SU8 washer?
Separately, the authors should discuss whether the graphene below the SU8 is also getting hydrogenated. Could the authors verify this by using Raman, for example by establishing a Raman map of the hydrogenation sites (within the resolution that Raman provides). What would be the LD between the hydrogenation sites (under the assumption LD could be calculated)?
4. The authors quantified how fast the hydrogenation occurs for a single device (e.g., the one reported in 1d). Have the authors performed experiments on more devices? How do the results vary across devices? Could the author comment on the number of devices they used for each experiment? Could the author provide an error analysis? I have a similar question about error analysis for the Fermi energy calculation in Figure 2 and for the hydrogenation onset potentials of corrugated graphene, graphene on hBN, and graphene on SiO₂ in Figure 3c? What is the mean and standard deviation for the hydrogenation peak currents and the electron currents?
5. The electrochemical reactions occurring on PdHx are also key to the mechanisms of graphene hydrogenation. In previous work (reference1), PdHx is used as a reference electrode together with a working electrode and a counter electrode (a 3-electrode electrochemical system). Is PdHx here the working, reference or counter electrode? And, as an electrode, what are the specific reaction equations that occur on PdHx when gaining and losing electrons?

6. Raman of hydrogenated graphene. In this work, hydrogenated graphene shows a D peak and a G peak with a ratio of 1:1. However, in previous reports on hydrogenated graphene^{2,3} Raman showed a D' peak next to the G peak, and the intensity ratio of D/G varied with the degree of hydrogenation. Did the authors also observed a D' peak? Is there an intermediate hydrogenation state when the D/G ratio is not 1:1?

7. Figure 3c shows the electronic current of graphene on hBN, on SiO₂ and for corrugated graphene. Could the authors also show the Dirac plots where the source drain current is plotted against VG (i.e., plot the electronic current vs VG-VNP over the entire voltage range starting from -2.2V to above +1V)? Highly corrugated graphene should exhibit minimal electron/hole mobility. Now the authors plot the source-drain current only in the region of VG-VNP < -1V in Figure 3c.

8. Could the electrolyte get underneath the graphene through the cracks (if any) and intercalation via the edges? If the electrolyte get underneath the graphene, are there any protons transporting through graphene to the electrolyte below it? Would graphene get hydrogenated underneath the SU8? In the reference⁴, the authors used a similar circuit design but measured the proton current across graphene rather than the hydrogenation current; see Fig. 1a when V_b = 0 and Fig. 1b in reference⁴). Are the results from ref 4 and those reported here consistent? Could protons also transport through graphene in this work? (i.e. in reference⁴, both graphene and the bottom channel electrodes share the same ground when V_b = 0, similarly to this work).

9. The cleanness of graphene electrodes in electrochemical experiments is very important. I believe that the best practice would be to grow graphene in situ in the electrochemical measurement glassware and after growth to incubate it with electrolytes without any transfer. Could the author comments on the influence of any residues that may appear from the device fabrication and the possible impact of these residues on the electrochemistry?

References:

1. Smith, G., Zalitis, C. M. & Kucernak, A. R. J. Thin solid state reference electrodes for use in solid polymer electrolytes. *Electrochem commun* 43, 43–46 (2014).
2. Li, S. et al. Large transport gap modulation in graphene via electric-field-controlled reversible hydrogenation. *Nat Electron* 4, 254–260 (2021).
3. Elias, D. C. et al. Control of Graphene's Properties by Reversible Hydrogenation: Evidence for Graphane. *Science* (1979) 323, 610–613 (2009).
4. Tong, J. et al. Control of proton transport and hydrogenation in double-gated graphene. *Nature* 630, 619–624 (2024).

Reviewer #2

(Remarks to the Author)

Reviewer #3

(Remarks to the Author)

The manuscript by Y.-C. Soong et al reports a study of the electrochemical hydrogenation of graphene and the consequences of this process on the transport properties of the material. The work roots in a previous publication in *Nature* 630, 619 (2024) by the same research groups. In the present article, the authors aim at providing further insights into the hydrogenation process by considering the effect of the replacement of hydrogen with deuterium and of the graphene roughness.

In general, the article appears rather narrow in terms of broad interest and devoted mainly to those readers with a specific technical expertise. The main results of the work, as well as their significance, are not always clear and several parts of the manuscript are not well described. In particular, the importance and role of the theoretical support is not easily understandable.

More specific criticisms and remarks follow.

In the “main” section on page 1 “has been since been established” should be corrected.

The description of the experimental set-up (e.g., suppl. Fig. 2 and Fig. 1b) seems rather incomplete. For instance, the PdHx electrodes are not indicated (is the topmost electrode in Fig. 1b?). In addition, the electrochemical current, its flow, sign, and significance are not adequately introduced.

The theoretical explanation provided at end of page 2 and beginning of page 3 is very hard to follow both by its content and by its necessity. The presentation of the results could proceed without that part that contributes making the article a bit confusing.

Are the data shown in Fig. 1a the same as those already presented in *Nature* 630, 619 (2024)? Or do they differ in some detail?

The red and blue arrows in Figure 1c are meant to be consistent with each other. However, the figure misses this purpose. The starting point of the process should be clearly indicated for both the electronic and electrochemical current, as well as the different phases of process occurring concomitantly should be adequately indicated by the arrows. In the figure caption, in the sentence “Black arrows mark the proton reduction process its oxidative desorption” an “and” is missing.

The caption of Fig. 1c is very confusing. It starts from the description of the bottom panel, then goes to the description of the “top-axis” but it doesn't describe the top panel, then it describes the “Inset” (there are many insets, it has to be specified

which one it is described) and finally it describes the top panel and the top insets. The fact that the bottom inset is described after the “top-axis” is very confusing, the caption should be re-organized.

The inset with the Raman data is seemingly compressed vertically, and the labels are written with very small characters. Being small and compressed, the labels cannot be read clearly.

The “exponential rise truncated at the insulating transition” mentioned at page 4 should rather be an exponential decay since the insulating transition occurs at low current values.

In general, the description of Fig. 1c is not sufficiently detailed and/or clear despite it is a pivotal result of the present work. The Raman results (Fig. 1 and Supplementary Fig. 8) are not so clear and are not discussed in detail. Why are the spectra of the pristine sample (or de-hydrogenated/deuterated) characterized by a much lower signal to noise ratio? Furthermore, why does the G peak show a much lower intensity than the 2D peak (up to disappearing almost completely, as shown in Supplementary Fig. 8) in the pristine samples at variance with the hydrogenated/deuterated ones? And why is there such a huge variation in the peak width for both the G and 2D modes? In Supplementary Fig. 8, around 1300 cm^{-1} , there are some “negative” spikes whose intensity is comparable to that of the G peak. What are they due to? It looks like the signal is very low and the spectra were not integrated much. Why instead are the data in Fig. 3b much better (i.e., with a much higher signal to noise ratio)? Fig 3c is very small, but it seems that the width of the G peak is not as narrow as in the other figures and it is maybe consistent with the width of the G peak shown for the hydrogenated/deuterated samples in Figs. 1 and 2.

The right panel in Supplementary Fig. 8 shows a sudden change in the Raman spectrum, from the absence of the D band to its presence. What happens in between? Can the hydrogenation/deuteration process be monitored? And how precise is the information which one can get from the Raman spectrum? On page 4, it is said that a deuterium concentration of $\sim 10^{14}\text{ cm}^{-2}$ is deduced by the Raman spectrum. How precise is this concentration value?

Generally, information on the integration time and total intensity of the data should be provided (whenever normalized spectra are shown, it would be useful to say how large would the intensity factor be for hydrogenated/deuterated and pristine samples).

Not enough information on the Raman measurements is provided (laser wavelength, information on the setup and detectors, etc.).

In Fig.2, it would be better to re-use for protons the same colors used in Fig. 1, e.g., red. This would also make the Figure clearer (the grey lines are not so well visible).

The effects of the morphology on the electronic/electrochemical currents seems the most interesting result of this work. The way it is presented is though misleading. On one hand, the reader gets the impression that corrugated graphene is beneficial, but from our understanding it is instead detrimental because a larger electrochemical current is required. On the other hand, corrugated graphene shows a higher electronic current that seems to point to a beneficial effect. This part of the work should be better explained as to remove this seemingly inconsistency.

In conclusion, the manuscript requires important revisions, and, in particular, it is not of adequate general interest for Nature Communications, but it is more suitable for a specialised journal.

Reviewer #4

(Remarks to the Author)

Version 1:

Reviewer comments:

Reviewer #1

(Remarks to the Author)

We thank the authors for their responses. We still have several questions after the authors have revised their manuscript.

1) Thanks to the authors for redrawing the circuit, which makes it clearer. The authors used a source meter (Keithley 2636B source meter, two-wire measurement) and a voltmeter (Keithley 2182A Nanovoltmeter) instead of using a potentiostat to perform CV scans. We believe that this platform, particularly the reference electrode, cannot provide feedback control for the current loop. The circuits in the references 1,2 seem to also use a source meter with a voltmeter. We therefore still question whether the use of a potentiostat would yield different results than using two separate Keithley instruments for determining hydrogenation overpotentials. In electrochemistry we use potentiostat as standard equipment.

2) In their response to point 2 the authors mention that the conductivity of deuterons in aqueous electrolytes is $\sim 10\%$ smaller than for protons (Arcis et). According to the CRC Handbook of Electrochemistry and Physics 3 (page 5-77), the conductivity of H^+ is 1.4 times that of D^+ in water at 25°C .

3) Separately, in Figure S6, the authors could not measure a significant difference of conductivity between H^+ and D^+ in HTFSI and DTFSI for SiN_x substrates with a hole without graphene, and attributed this result to device-to-device variations (see point 2, where a difference of 1.4 is reported for bulk conductivities of H^+ and D^+). Therefore, we question whether

PdHx and PdDx are reliable in detecting proton or deuteron currents.

4) In our initial question we asked which electrochemical reactions occur on PdHx. We asked this question because we believe that the electrochemical reactions occurring at the reference electrode are key to determine the overpotential of hydrogenation on graphene. We asked the following question: "what are the specific reaction equations that occur on PdHx when gaining and losing electrons?" In their revised manuscript, the authors mention that "Pd + H⁺⁺ e⁻ → PdHx" is happening. What if the reaction would be "Pd + xH⁺ + xe⁻ → PdHx. Separately, if PdHx is saturated with H, atomic hydrogen or hydrides can no longer be inserted into the electrode. The literature report on an upper limit to the electrochemical insertion of H into Pd, which could be considered here for the discussion.⁵ Possible reactions include 6:(see pdf)

5) Overall and over the time of the measurement, the composition of the reference electrode may change and therefore such reference electrode may not be used to determine the overpotential of hydrogenation on graphene. In order determine whether the reference electrode composition changes (or not), the authors could use XRD or XPS before and after the measurement.

6) In the lower panel of figure 3C, corrugated graphene shows a higher current compare to graphene on hBN and graphene on SiO₂; the authors carried out an error analysis and concluded that differences in currents are attributed to device-to-device variations (now the author normalized the electronic current to the width of the graphene). We expect corrugated graphene to have lower charge carrier mobilities than less corrugated graphene. What if the authors would calculate and compare the carrier mobility for the different samples instead of the current^{7,8}; and establish whether corrugated graphene has a better affinity for H⁺ (see also point 5).

7) Corrugated graphene should have much lower overpotential of hydrogenation comparing to graphene on SiO₂ and hBN. However, corrugated graphene was measured to have the similar overpotential as for graphene on SiO₂ and hBN (the potential at the peak of hydrogenation current in upper panel of Figure 3c). Similarly, corrugated graphene should have much lower overpotential at the insulating transition compared to graphene on SiO₂ and hBN. However, corrugated graphene was measured to have the maximum overpotential (the potential at the zero current transition in lower panel of Figure 3c) compare to graphene on SiO₂ and hBN. The authors explain these results as "concave areas relatively free of protons [...] leaves electronically conductive paths". However, this reason is unclear. Drawing a schematic diagram of electronically conductive paths in the concave and convex areas may be relevant. In addition, if the concave areas with conductive paths exist, why does corrugated graphene undergo the same insulating transition as graphene on SiO₂ and hBN?

8) The size of the washer compared to the size of graphene is very important for the fully insulation transition. So, we asked before, "Are the graphene samples studied here bigger or smaller than the opening in the SU8 washer?" The authors point out that the graphene flake must be smaller than the opening of the SU8 washer, which is important for the insulating transition. We strongly agree with this point of view. However, in a previous paper by the same research group¹, the graphene flake seems larger than the opening in the SU8 washer, and an insulating transition is also observed. Are there here any contradictions between the two situations/articles?

References

1. Tong, J. et al. Control of proton transport and hydrogenation in double-gated graphene. *Nature* 630, 619–624 (2024).
2. Cai, J. et al. Wien effect in interfacial water dissociation through proton-permeable graphene electrodes. *Nat Commun* 13, 5776 (2022).
3. CRC Handbook of Chemistry and Physics. (CRC Press, 2014). doi:10.1201/b17118.
4. Lozada-Hidalgo, M. et al. Sieving hydrogen isotopes through two-dimensional crystals. *Science* (1979) 351, 68–70 (2016).
5. Benck, J. D., Jackson, A., Young, D., Rettenwander, D. & Chiang, Y.-M. Producing High Concentrations of Hydrogen in Palladium via Electrochemical Insertion from Aqueous and Solid Electrolytes. *Chemistry of Materials* 31, 4234–4245 (2019).
6. Moumaneix, L., Rautakorpi, A. & Kallio, T. Controlling the Reactivity and Interactions between Hydrogen and Palladium Nanoparticles via Management of the Particle Diameter. *ChemElectroChem* 10, (2023).
7. Zhang, Z., Xu, H., Zhong, H. & Peng, L.-M. Direct extraction of carrier mobility in graphene field-effect transistor using current-voltage and capacitance-voltage measurements. *Appl Phys Lett* 101, (2012).
8. Zhong, H., Zhang, Z., Xu, H., Qiu, C. & Peng, L.-M. Comparison of mobility extraction methods based on field-effect measurements for graphene. *AIP Adv* 5, 057136 (2015).

Reviewer #2

(Remarks to the Author)

Reviewer #3

(Remarks to the Author)

The authors replied to the previous concerns and the manuscript has improved, yet some aspects regarding Raman measurements remain not fully clarified, such as the quality of the spectra. In particular, while it has now been clarified that the spectra of the pristine samples are affected by the presence of an electrolyte background signal, this should not impede to have a better signal-to-noise ratio. In the methods section, it is mentioned that for Raman measurements, "accumulating time of 75s over 4 cycles" were used. Were both the pristine and hydrogenated sample spectra taken under these same conditions?

In the caption of figure 1, it is said "Raman spectra of pristine and hydrogenated samples. The background signal from the electrolyte was subtracted; the latter spectrum was divided by 5 for clarity". From this, it seems that there is only a factor of 5 in the intensity of the spectra, while the signal-to-noise-ratio seems to be lower than that by looking at the spectra.

Furthermore, the spectra in Fig. 3b look better than those in Fig. 1 and Supplementary Fig. 8. Is this discrepancy due to different accumulation times or to something else?

As for Figure 1, we know that the data of Figure 1 are different to those shown in 2024 Nature paper. In our previous report, we were referring to Fig. 1a (now Fig. 1d) showing the Energy vs distance plot. This plot shows the same data plotted in Extended Data Fig. 8 (zero electric field). We are sorry we were not clear enough.

This said, while we believe that the manuscript was improved, we still feel that this work would be more suitable for a specialised journal.

Reviewer #4

(Remarks to the Author)

Version 2:

Reviewer comments:

Reviewer #1

(Remarks to the Author)

Below are my detailed comments and suggestions, which may be helpful for revising the manuscript:

1. Circuit reproducibility and experimental setup

We thank the authors for elaborating on their circuit and introducing a discussion on using the Keithley 2636B as a potentiostat. However, the current explanation remains ambiguous. It would significantly improve clarity and reproducibility if the authors could provide a schematic diagram showing how the Keithley 2636B sourcemeter and the 2182A nanovoltmeter are connected to the graphene and PdHx electrodes. This would assist future researchers in reproducing the setup accurately. Still, I would not be surprised that doing similar experiments with a potentiostat would yield different onset potentials for hydrogenations and dehydrogenations and therefore results/interpretations/conclusions.

Additionally, could the authors clarify whether the Keithley 2636B sourcemeter was operated in 2-wire or 4-wire mode? As the manufacturer's manual notes, the sourcemeter functions as a potentiostat only in 4-wire mode, where the Sense LO provides feedback to Force LO. If used in 2-wire mode, this feedback loop is not valid, which would compromise the reliability of the measured potentials at the reference electrode.

see Figure 2 of this manual on the website of Tek: <https://www.tek.com/en/documents/application-note/keithley-instrumentation-electrochemical-test-methods-and-applications>

2. Use a potentiostat to perform CV on graphene hydrogenation

While we appreciate the CV measurements provided for the Pt electrode in a $K_3Fe(CN)_6$ solution (Fig R1.1 of the rebuttal), this alone, without a counter electrode does not validate the electrochemical circuit used in the manuscript. If the authors aim to demonstrate the validity of their custom circuit, they could consider performing measurements using a commercial potentiostat on graphene hydrogenation with PdHx electrodes and compare the results obtained with their setup using the two Keithleys.

3. Ambiguity in proton and deuteron transport

The manuscript should clarify the differences observed between H⁺ and D⁺ transport in free pore measurements in the context of mobility similarities/differences in HTFSI/PEG systems or whether those differences may arise from the electrode–electrolyte contact resistance. A more detailed discussion or additional control experiments would help clarify the interpretation of the data.

4. PdHx electrode characterization and proton transport in HTFSI/PEG

The PdHx electrode is used for detecting hydronium ion currents, as demonstrated in prior literature involving hydrated media (e.g., maleic–chitosan, ref 1). However, PEG is not hydrated. There must be some difference between hydronium ion and HTFSI when those interact with PdHx. The manuscript should carefully discuss how proton exits in HTFSI and how they react with PdHx electrodes.

Moreover, it is unclear how the authors prepared the PdHx electrode. In previous studies, the PdHx electrode were saturated with hydrogen (ref 1–3). If saturation occurred, the ability of PdHx to absorb further protons would be limited. Could the author describe more in details how they prepared the PdHx electrodes.

Additionally, the electrochemical equation on page 7 appears unbalanced. A more rigorous form would be: $xH^+ + xe^- + Pd \rightarrow PdH_x$. This should be corrected for clarity and chemical accuracy.

5. Mobility estimation and interpretation of insulating behavior

The authors do not use two-end (source and drain) field-effect-based mobility analysis, despite its widespread adoption in similar studies (Ref 4–8). If there is a known inconsistency in applying this method to the present system, it should be discussed explicitly. Avoiding this issue weakens the interpretation of the observed insulating transition, which is based on carrier conduction but lacks support of quantitative carrier mobility and carrier concentration.

6. Role of curvature in hydrogenation

The authors emphasize that curvature (convex and concave regions) affects graphene hydrogenation, yet they don't provide direct visual or experimental support for this claim. Including schematics or Raman mapping (e.g., spatial or time/potential-resolved analysis of D-band changes) could help clarify whether such morphological effects exist and how they influence reactivity.

7. Novelty

Fig 3c shows a largely enhanced dehydrogenation current for corrugated graphene. This is the impressive part of the results. However, mechanistically, the hydrogenation of graphene has been reported in multiple prior studies (e.g., Refs.9–15). Several mechanistic discussions are also available in literature (e.g., Refs.16,17). It is still not clear from the manuscript and data what is the mechanism of hydrogenation and dehydrogenation.

Would the author measure similar high electrochemical currents in dry Nafion/PdHx systems (ref3)? Again I am not sure the electrochemistry is right in HTFSI (see above). Here also, I wonder whether switching to using a potentiostat would yield different results? However, I acknowledge that from Fig 3c top panel, the results are impressive.

1. Zhong, C. et al. A polysaccharide bioprotonic field-effect transistor. *Nat Commun* 2, 476 (2011).
2. Morgan, H., Pethig, R. & Stevens, G. T. A proton-injecting technique for the measurement of hydration-dependent protonic conductivity. *J Phys E* 19, 80–82 (1986).
3. Hu, S. et al. Proton transport through one-atom-thick crystals. *Nature* 516, 227–230 (2014).
4. Lv, H. et al. High carrier mobility in suspended-channel graphene field effect transistors. *Appl Phys Lett* 103, (2013).
5. Xu, H. et al. Top-Gated Graphene Field-Effect Transistors with High Normalized Transconductance and Designable Dirac Point Voltage. *ACS Nano* 5, 5031–5037 (2011).
6. Kim, S. et al. Realization of a high mobility dual-gated graphene field-effect transistor with Al₂O₃ dielectric. *Appl Phys Lett* 94, (2009).
7. Farmer, D. B. et al. Utilization of a Buffered Dielectric to Achieve High Field-Effect Carrier Mobility in Graphene Transistors. *Nano Lett* 9, 4474–4478 (2009).
8. Liao, L. et al. High- κ oxide nanoribbons as gate dielectrics for high mobility top-gated graphene transistors. *Proceedings of the National Academy of Sciences* 107, 6711–6715 (2010).
9. Elias, D. C. et al. Control of Graphene's Properties by Reversible Hydrogenation: Evidence for Graphane. *Science* (1979) 323, 610–613 (2009).
10. Li, S. et al. Large transport gap modulation in graphene via electric-field-controlled reversible hydrogenation. *Nat Electron* 4, 254–260 (2021).
11. Tong, J. et al. Control of proton transport and hydrogenation in double-gated graphene. *Nature* 630, 619–624 (2024).
12. Zhao, M., Guo, X.-Y., Ambacher, O., Nebel, C. E. & Hoffmann, R. Electrochemical generation of hydrogenated graphene flakes. *Carbon N Y* 83, 128–135 (2015).
13. Fei, Y., Fang, S. & Hu, Y. H. Synthesis, properties and potential applications of hydrogenated graphene. *Chemical Engineering Journal* 397, 125408 (2020).
14. Hu, Y. et al. Synaptic transistor based on reversible hydrogenation of graphene channel. *Appl Phys Lett* 126, (2025).
15. Cha, J., Choi, H. & Hong, J. Damage-free hydrogenation of graphene via ion energy control in plasma. *Applied Physics Express* 15, 015002 (2022).
16. Daniels, K. M. et al. Mechanism of Electrochemical Hydrogenation of Epitaxial Graphene. *J Electrochem Soc* 162, E37–E42 (2015).
17. Podlivaev, A. I. & Katin, K. P. Competition of hydrogen desorption and migration on graphene surface in alternating electric field: Multiscale molecular dynamics and diffusion study. *Appl Surf Sci* 686, 162125 (2025).

Reviewer #2

(Remarks to the Author)

Version 3:

Reviewer comments:

Reviewer #1

(Remarks to the Author)

Thanks for the reply. We appreciate the efforts of the authors to answer our questions and revise their manuscript. We found a few points in the revised files that merit some attention from the authors. After that the manuscript can be published.

(1) All the electron current data were replaced in the revised manuscript with conductance values. However, the plots have similar absolute values (i.e. conductance vs current). Is this correct (Figure 1b, Figure 2, bottom panel, Figure 3c bottom panel)? Taking into example Figure 1b: the absolute value now of the conductance is the same as the current they reported in the previous version. For that to be correct, length to width ratio of the graphene device should be strictly 1:1? However, in Figure S2b, the shape of the graphene is irregular and the gold electrode is a V-shaped arrangement. Similarly in the bottom panel of Figure 2, in the revised manuscript, the red and blue curves now touches at $V_g - V_{np} = -1.2V$ but not in the original manuscript. If the authors applied a scaling factor of 1.13, what is the reason? Or are the devices not at the same scale? For the point on the blue line where the current is 441.4 nA, the corresponding conductance value is 497.6 μS .

We have a similar question concerning the lower panel of Figure 3c: the vertical axis value for the Gr-SiO₂ sample remain unchanged when switching from current to conductance, while the vertical axis values of Gr-hBN and Corrugated Gr seems scaled by a certain factor? For example, the scaling factor of the Gr-hBN curve is 0.90, while the scaling factor of the Corrugated-Gr curve is 0.50. We do not understand the origin of these scaling factors reported now in the revised manuscript.

(2) The data point of the original Figure S6 for the 10 μm pore is not shown in the new plot (Figure S6) of the new manuscript. This data point shows that the conductance of DTFSI is 0.19 μS and the conductance of HTFSI is 0.17 μS . Including these data points in the fitting of circular hole etched in the SiN_x membrane would yield a larger bulk conductance for H and D.

We calculated those 0.19 μS using the intercepts: For DTFSI curve, we read two points :(-3, -0.56), (3, 0.57). The slope is $(0.57+0.56)/(3+3)=0.19$ (μS)

Similarly, for HTFSI curve, we read two points : (-3, -0.49), (3, 0.51). The slope is $(0.51+0.49)/(3+3)=0.17$ (μS)

(3) Concerning the hydrogen saturation of the PdH_x electrode: According to Murphy, D. W. et al, Chem. Mater., 1993, 5, 767-769, the Pd was immersed in a BH₄⁻ solution overnight while stirring. The previous study emphasized that the Pd was saturated with H; however, this article claims that Pd electrode has intermediate H loading. Why did the loading stop at intermediate state? In addition, if a Pd electrode that is already saturated with H, which means that it can only lose protons but cannot gain protons. Can it still be used as a reference electrode or counter electrode here?

(4) The calculation of the carrier mobility of Gr-BN. We proposed that the authors could calculate the carrier mobility for their three type of samples for comparison. Could the authors provide the value of Gr-BN as well?

Reviewer #2

(Remarks to the Author)

Version 4:

Reviewer comments:

Reviewer #1

(Remarks to the Author)

Thank you for the clarifications and answers. We do not have further comments and like to congratulate the authors for their work.

Reviewer #2

(Remarks to the Author)

I co-reviewed this manuscript with one of the reviewers who provided the listed reports. This is part of the Nature Communications initiative to facilitate training in peer review and to provide appropriate recognition for Early Career

Researchers who co-review manuscripts.

Response to comments from Reviewer #1

Referee report for: Mechanism of the electrochemical hydrogenation of graphene

In this manuscript, Y.-C. Soong and co-authors discussed the mechanism of the electrochemical hydrogenation of graphene with the acid electrolyte (HTFSI) dissolved in PEG. They found differences in oxidation and reduction currents during cyclic voltammetry and demonstrated that the hydrogenation proceeds as a reduction reaction in which proton adsorption competes with a process attributable to the formation of H₂ molecules. The work is really interesting and important. I have a few questions that trigger my curiosity.

We are very grateful with the Reviewer for this encouraging assessment of our work. We have done our best to clarify the points raised, which we believe have improved the clarity of the manuscript.

1. The circuit design involves several electrodes (Figure 1b and Supplementary Figure 2a). Are those all PdHx electrodes? Are there relationships between them, is there an electrochemical current running between them or could one electrode influence the other during the measurements (see attached figure)? It seems to me that there are now two reference electrodes and a bias is applied between the two. Are the gate and the reference electrode biased with respect to each other? In other words, if an electrochemical reaction occurs at the graphene surface and that the graphene is used as a GFET, how to disentangle the two contributions (i.e., electrochemistry and field effect)?

The setup schematics in Fig. 1 and Fig. S2 were not drawn clearly, which we believe is the source of the confusion. We have redrawn them in the revised manuscript.

In brief, the devices are connected to two PdHx electrodes. One functions as gate electrode and the other as a reference electrode. The current flowing into the reference electrode is zero and there is no voltage applied between these two electrodes. The setup is similar to that reported in ref. 15 and by Y. He et al, Nat. Materials 2019, 22, 1098-1104. The voltage V_G is the potential of graphene vs the reference electrode. We found that this potential is effectively the same as the applied gate potential, within an error of <4 mV, similar to our findings in ref. 15 (Extended Data Fig. 2 in that reference). Nevertheless, we decided to keep the reference electrode, as most electrochemical data is reported this way. We added this discussion to the section 'Electrical and electrochemical measurements' in Methods section, page 8.

2. Could the authors discuss the mechanism of proton migration in the electrolyte (Grotthuss mechanism, or vehicle mechanism). Supplementary Figure 6 shows that the conductivity of proton (in HTFSI) is smaller than that of deuteron (in DTFSI). However, it is known that the mobility of proton should be greater than that of deuteron because the mass difference. We observed the same electrolyte conductivity with HTFSI and DTFSI electrolytes within our experimental resolution, which is hardly surprising. The conductivity of deuterons in aqueous electrolytes is typically ~10% smaller than for protons (e.g. H. Arcis, et al. Phys Chem B 2022, 126, 8791-8803), and the same can be expected for DTFSI. This relatively small difference is within our typical device-to-device variability for control devices without graphene. We have clarified this point in the caption in Fig. S6.

Regarding the proton transport mechanism in the electrolyte, there is much less literature for HTFSI than for aqueous electrolytes. However, most references we found suggest that proton transport is primarily via a Grotthuss mechanism (A. A. Moses & C. Arntsen Phys Chem Chem Phys 2023, 25, 2142 & M. L. Hoarfrost et al. J. Phys Chem B 2012, 116, 8201-8209).

3. In the manuscript, the graphene located in the SU8 washer opening is getting hydrogenated.

Figure S2 suggests that the graphene flake is smaller than the opening in the SU8. Are the graphene samples studied here bigger or smaller than the opening in the SU8 washer? Separately, the authors should discuss whether the graphene below the SU8 is also getting hydrogenated. Could the authors verify this by using Raman, for example by establishing a Raman map of the hydrogenation sites (within the resolution that Raman provides). What would be the LD between the hydrogenation sites (under the assumption LD could be calculated)? The graphene flakes are slightly smaller than the opening in the SU8 mask. This is because observing the insulating transition requires all electronic conduction paths along the graphene flake to be closed. This in turn requires a full section of the flake to be hydrogenated. This will not be possible if the flake is larger than the SU8 opening because the areas covered with SU8 are not hydrogenated. This is consistent with previous control experiments in which the areas covered by the SU8 are probed electronically and are found to be not hydrogenated (e.g. Fig. 2 in ref. 14 in the manuscript). We have included this discussion in the section ‘Device fabrication’ in Methods, page 7.

Regarding the Raman, from the ratio of intensity of the D to G bands in hydrogenated samples, $I_D/I_G \approx 1$, we can estimate that the distance between H atoms in graphene is $L_D \approx 1$ nm. This corresponds to a concentration of H atoms of $\approx 1 \times 10^{14} \text{ cm}^{-2}$. Estimates of defect density using the Raman spectra are accurate within order of magnitude (Luchese et al Carbon 2010, 48, 1592-1597 and Jorio et al, Physica Status Solidi B 2010, 247, 2980-2982). However, the electronic transport measurements find that the hydrogenation transition takes place for a charge carrier density of $\approx 1 \times 10^{14} \text{ cm}^{-2}$ (the Fermi energy extracted from these measurements can be readily converted into charge density) in agreement with the Raman experiments. Hence, the concentration of H atoms in the hydrogenated phase can be estimated with a fair degree of accuracy. We have included an expanded discussion of the Raman experiments in the Methods section, page 8.

4. The authors quantified how fast the hydrogenation occurs for a single device (e.g., the one reported in 1d). Have the authors performed experiments on more devices? How do the results vary across devices? Could the author comment on the number of devices they used for each experiment? Could the author provide an error analysis? I have a similar question about error analysis for the Fermi energy calculation in Figure 2 and for the hydrogenation onset potentials of corrugated graphene, graphene on hBN, and graphene on SiO₂ in Figure 3c? What is the mean and standard deviation for the hydrogenation peak currents and the electron currents? All experiments in this work were carried out with at least 3 and up to 8 devices per experiment. We found small variability from device to device. Below we specify the statistics for the different quantities requested and the sample number for each of the experiments. This information is now in Supplementary Table 1 and in the text.

(a) Hydrogenation potentials in Fig. 3c

Gr-on-SiO ₂ , -1.75 ± 0.08 V	8 devices
Corrugated Gr, -2.08 ± 0.13 V	4 devices
Gr on BN, -1.99 ± 0.02 V	4 devices

(b) Hydrogenation peak currents in Figure 3c (normalised by exposed flake area)

Gr on SiO ₂ , 0.19 ± 0.05 mA cm ⁻²	8 devices
Corrugated Gr, 4.3 ± 0.94 mA cm ⁻²	4 devices
Gr on BN, 0.23 ± 0.02 mA cm ⁻²	4 devices

(c) Maximum electronic current in Figure 3c (normalised by the minimum width of the exposed flake section)

Gr on SiO ₂ , 49 ± 10.4 mA m ⁻¹	8 devices
Corrugated Gr, 41 ± 12.3 mA m ⁻¹	4 devices
Gr on BN, 45 ± 7.4 mA m ⁻¹	4 devices

- (d) Fig. 1e in the revised manuscript shows the reaction rate as a function of applied potential from three different devices. The scatter is ~10%.
- (e) Statistic for the Fermi energy calculation in Figure 2. Onset potential for deuteration is 0.82 ± 0.015 eV and 0.84 ± 0.005 eV for hydrogenation. 8 devices for hydrogenation and 4 for deuteration experiments.

5. The electrochemical reactions occurring on PdHx are also key to the mechanisms of graphene hydrogenation. In previous work (reference1), PdHx is used as a reference electrode together with a working electrode and a counter electrode (a 3-electrode electrochemical system). Is PdHx here the working, reference or counter electrode? And, as an electrode, what are the specific reaction equations that occur on PdHx when gaining and losing electrons?

Our setup is equivalent to a three-electrode electrochemical cell in which both counter and reference electrodes are PdHx. The reaction taking place in the counter and reference electrode is proton adsorption into the metal: $H^+ + e^- + Pd \rightarrow PdHx$ (e.g. M. Grden, et al. Electrochimica Acta 2008, 53, 7583-7598). This is an important property of Pd that is exploited when using this metal as either pseudo reference electrode or to reduce contact resistance in nano-electronic devices working with protons (e.g. A. J. Tan, et al. Nature Materials 2019, 18, 35-41 or C. Zhong et al. Nature Communications 2011, 2:476). This discussion has been added to the section 'Device fabrication' in Methods, page 7.

6. Raman of hydrogenated graphene. In this work, hydrogenated graphene shows a D peak and a G peak with a ratio of 1:1. However, in previous reports on hydrogenated graphene^{2,3} Raman showed a D' peak next to the G peak, and the intensity ratio of D/G varied with the degree of hydrogenation. Did the authors also observed a D' peak? Is there an intermediate hydrogenation state when the D/G ratio is not 1:1?

Fig. R1.1. (a) Analysis of Raman spectra of hydrogenated samples. Data points, experimental observation. Red, green and blue curves are fitted Lorentzian for D, G and D' bands, respectively. Black curve, Sum of D, G and D' bands. I_D/I_G and A_D/A_G are the intensity and integrated-area ratios for the D and G bands, respectively. **(b)** Raman spectra of samples during a typical dehydrogenation cycle. Numbers 1-5 mark the sequence of measurements. Starting from -1.5 V vs NP (spectrum #1), the 2D band is initially smeared. At -1V, the G band starts becoming less prominent. At 0V, the 2D band becomes sharp, but the D band is still apparent. At 0.7 V, the

spectrum of pristine graphene is recovered, but the device is doped due to the applied voltage, yielding a relatively less sharp 2D band. Finally (spectrum #5), the voltage is swept back to 0V vs NP, yielding the spectra of pristine, undoped graphene. These findings are very similar to those in Fig. 1, ref. 14.

Fig. R1.1a shows analysis of a typical Raman spectra for hydrogenated graphene samples. This shows that D band is convoluted with the large G band.

Regarding the second question, we did not observe intermediate phases during the hydrogenation of graphene in the Raman spectra. We observe the sudden appearance of a sharp D and G band. This is consistent with the electronic measurements, which show a sharp conductor insulator transition. Even at moderate bias (e.g. -1.5 vs NP), graphene becomes hydrogenated in a couple of seconds (as shown in Fig. 1e).

This finding is also consistent with ref. 14. The authors of that work observed intermediate hydrogenated states, but only during the de-hydrogenation process. We also observe this. Fig. R1.1b shows that the Raman spectra evolve from the highly disordered phase to pristine graphene more gradually.

7. Figure 3c shows the electronic current of graphene on hBN, on SiO₂ and for corrugated graphene. Could the authors also show the Dirac plots where the source drain current is plotted against VG (i.e., plot the electronic current vs VG-VNP over the entire voltage range starting from -2.2V to above +1V)? Highly corrugated graphene should exhibit minimal electron/hole mobility. Now the authors plot the source-drain current only in the region of VG-VNP < -1V in Figure 3c.

Fig. R1.2 shows the electronic response over the full voltage range for the devices in Fig. 3, normalised by the width of the graphene flake. The figure shows that there are no significant differences in the electronic current between the different devices (corrugated or supported on hBN and SiO₂) within the expected device-to-device scatter reported in point #4, when normalised by flake dimensions. This is consistent with previous works. Note that while highly corrugated graphene should display a high density of electron hole puddles and charge localisation (e.g. e.g. J. Martin, et al. Nature Physics 2008, 4, 144-148 and ref. 34), it is not expected to display minimal electron/hole mobility at room temperature. This is because the length scale of corrugations is larger than the Fermi wavelength of charge carriers in graphene for the Fermi level in our devices. To observe the effect of charge localisation in electronic transport measurements, it is necessary to go to Fermi energy levels of the order $\sim 10^{10} - 10^{11} \text{ cm}^{-2}$ (e.g. J. Martin et al), which are inaccessible at ambient conditions.

Fig. R1.2. Electronic current from the graphene-on-silicon oxide (blue), graphene-on-hBN (green) and corrugated graphene (red) samples shown in Fig. 3 in the main text. The difference in electronic current between data sets is comparable to device-device variation within the same type of device (see point #4).

8. Could the electrolyte get underneath the graphene through the cracks (if any) and intercalation via the edges? If the electrolyte get underneath the graphene, are there any protons transporting

through graphene to the electrolyte below it? Would graphene get hydrogenated underneath the SU8? In the reference⁴, the authors used a similar circuit design but measured the proton current across graphene rather than the hydrogenation current; see Fig. 1a when $V_b = 0$ and Fig. 1b in reference⁴). Are the results from ref 4 and those reported here consistent? Could protons also transport through graphene in this work? (i.e. in reference⁴, both graphene and the bottom channel electrodes share the same ground when $V_b = 0$, similarly to this work).

We did not observe electrolyte intercalation between graphene and the substrate, consistent with the strong van der Waals forces between 2D crystals and oxide surfaces (e.g. S. Koenig, et al Nature Nanotechnology 2011, 6, 543-546).

Regarding comparison with our previous work, in Tong et al we worked with suspended graphene devices. In that case, the electrochemical signal was dominated by proton transport through graphene, which yields large currents of the order of 10 mA cm^{-2} , even for zero back gate voltage. In contrast, the present devices are supported on substrates (e.g. SiO_2) and display very small electrochemical currents of the order of 0.1 mA cm^{-2} . The shape of the I-V response is also completely different in the two works. In Tong et al. there is a continuous rise of current with applied voltage, whereas in the present work the current is negligible except at the hydrogenation and dehydrogenation potentials. Both works are consistent. In the present work there is no proton transport through graphene because transferred protons have no way of closing the electrochemical circuit. Conversely, our understanding is that the hydrogenation current is also present in ref. 15. but, because it is so small compared to the huge proton transport signal, it is not discernible in the experiments.

9. The cleanness of graphene electrodes in electrochemical experiments is very important. I believe that the best practice would be to grow graphene in situ in the electrochemical measurement glassware and after growth to incubate it with electrolytes without any transfer. Could the author comments on the influence of any residues that may appear from the device fabrication and the possible impact of these residues on the electrochemistry? Growing graphene via CVD in the target substrate can produce high quality samples. However, one can never rule out the presence of defects, such as vacancies or grain boundaries in CVD graphene, which can have significant impact on electrochemical process and thus complicates the interpretation of the data. To avoid this problem, most groups studying fundamental phenomena in graphene work with mechanically exfoliated crystals, which are defect free. To ensure the cleanliness of the exfoliated crystals, we use transfer recipes that have been optimised through many years and are widely reported in the literature. This yields very clean samples that can be studied in high precision electronic transport experiments or, as we showed recently, in electrochemical measurements with nanoscale spatial resolution and fA current resolution (e.g. O. J. Wahab, et al. Nature 620, 782-786, 2023, ref. 31 in manuscript).

References:

1. Smith, G., Zalitis, C. M. & Kucernak, A. R. J. Thin solid state reference electrodes for use in solid polymer electrolytes. *Electrochem commun* 43, 43–46 (2014).
2. Li, S. et al. Large transport gap modulation in graphene via electric-field-controlled reversible hydrogenation. *Nat Electron* 4, 254–260 (2021).
3. Elias, D. C. et al. Control of Graphene's Properties by Reversible Hydrogenation: Evidence for Graphane. *Science* (1979) 323, 610–613 (2009).
4. Tong, J. et al. Control of proton transport and hydrogenation in double-gated graphene. *Nature* 630, 619–624 (2024).

Response to comments from Reviewer #2

I co-reviewed this manuscript with one of the reviewers who provided the listed reports. This is part of the Nature Communications initiative to facilitate training in peer review and to provide appropriate recognition for Early Career Researchers who co-review manuscript

We thank the Reviewer for taking the time to look at our manuscript.

Response to comments from Reviewer #3

The manuscript by Y.-C. Soong et al reports a study of the electrochemical hydrogenation of graphene and the consequences of this process on the transport properties of the material. The work roots in a previous publication in Nature 630, 619 (2024) by the same research groups. In the present article, the authors aim at providing further insights into the hydrogenation process by considering the effect of the replacement of hydrogen with deuterium and of the graphene roughness.

In general, the article appears rather narrow in terms of broad interest and devoted mainly to those readers with a specific technical expertise. The main results of the work, as well as their significance, are not always clear and several parts of the manuscript are not well described. In particular, the importance and role of the theoretical support is not easily understandable. More specific criticisms and remarks follow.

The electrochemical hydrogenation of graphene has been recently shown to drive a robust and reversible conductor-insulator transition, which is of strong interest in computing applications based on ionic-electronic processes. However, the mechanism for this new process remains unknown. Glaring gaps in knowledge remain, such as why this reaction happens at the potentials observed, or even what is the reaction rate of this process as a function of potential. This is a problem, because processes that are not understood cannot be fully controlled in applications. Our work reports the electrochemical fingerprint of the hydrogenation reaction for the first time (Fig. 1a). This allowed us to elucidate the mechanism for this process and answered these open questions. It also addresses the fundamental question of how a proton adsorbs electrochemically to a pristine sheet of carbon atoms. These findings are of general importance to communities working on electrochemistry, solid-liquid interfaces and ion-based computing; and their importance was highlighted by Reviewer #1.

We have carefully addressed all the points raised by the Reviewer, which we believe have improved the clarity of our manuscript.

In the “main” section on page 1 “has been since been established” should be corrected.

We have rephrased this statement.

The description of the experimental set-up (e.g., suppl. Fig. 2 and Fig. 1b) seems rather incomplete. For instance, the PdHx electrodes are not indicated (is the topmost electrode in Fig. 1b?). In addition, the electrochemical current, its flow, sign, and significance are not adequately introduced.

We have clarified the schematic for the experimental setup in Fig. 1 and Fig. S2. Regarding the electrochemical current, protons flow from the gate electrode to graphene during the reduction process (starting at about -1.5 vs NP), yielding a negative current. Conversely, protons flow from graphene to the gate electrode during the oxidation process (at about 0 vs NP) and yield a positive current. The significance of these currents is discussed in detail in the section ‘Electrochemical response’ in the main text.

This discussion has been added in the section ‘Electrical and electrochemical measurements’ in Methods, end of first paragraph, page 8.

The theoretical explanation provided at end of page 2 and beginning of page 3 is very hard to follow both by its content and by its necessity. The presentation of the results could proceed without that part that contributes making the article a bit confusing.

Following the Reviewer comment, we have moved the theory further down in the main text to a new section titled ‘Electrochemical potentials’ and simplified the discussion as much as possible. However, we respectfully disagree with the Reviewer on the point that this theory is not necessary. Understanding why the electrochemical hydrogenation of graphene takes place at the potential observed is essential to reconcile the experimental observations with DFT predictions. DFT predicts a low energy barrier of ~0.3 eV, which differs markedly with the experiments because these observe this process when graphene’s Fermi level is about ~1 eV. This difference has been now reconciled using standard thermodynamic theory that calculates the Gibbs energy of the process based on the previous DFT calculations. This is an approach routinely used to analyse electrochemical data.

Are the data shown in Fig. 1a the same as those already presented in Nature 630, 619 (2024)? Or do they differ in some detail?

The electrochemical data in Fig. 1 is different to that in Tong et al. (Nature 2024) in qualitative shape, magnitude, the device structure it comes from and the underlying electrochemical process. In Tong et al, we studied suspended graphene devices. The electrochemical signal is dominated by proton transport through graphene, which yields currents that continuously rise with voltage and are of the order of 10 mA cm⁻². In contrast, the present work studies graphene supported on substrates, which do not allow for proton transport through graphene. In this case, the electrochemical response is dominated by the hydrogenation reaction on the graphene surface, yielding currents that are only significant at the hydrogenation and dehydrogenation transitions. Even then, the currents are only of the order of 0.1 mA cm⁻². Hence, the electrochemical response of the devices in this work are fundamentally different to those in Tong et al. Crucially, the new data in this manuscript reports the electrochemical fingerprint of the hydrogenation reaction, which had not been reported before and without which it is not possible to elucidate its mechanism.

The electronic current in both works is naturally the same and is presented here only to reference the processes behind the peaks in the electrochemical response. Following the Reviewer comment, this has been stressed in the main text, at the beginning of the section ‘Electronic response’.

The red and blue arrows in Figure 1c are meant to be consistent with each other. However, the figure misses this purpose. The starting point of the process should be clearly indicated for both the electronic and electrochemical current, as well as the different phases of process occurring concomitantly should be adequately indicated by the arrows. In the figure caption, in the sentence “Black arrows mark the proton reduction process its oxidative desorption” an “and” is missing.

Following the Reviewer comment, we have added numbers in both electronic and electrochemical curves to make the comparison clearer. This was a helpful suggestion.

The caption of Fig. 1c is very confusing. It starts from the description of the bottom panel, then

goes to the description of the “top-axis” but it doesn’t describe the top panel, then it describes the “Inset” (there are many insets, it has to be specified which one it is described) and finally it describes the top panel and the top insets. The fact that the bottom inset is described after the “top-axis” is very confusing, the caption should be re-organized.

We have relabelled the different panels in the graph and have reorganised the caption for clarity.

The inset with the Raman data is seemingly compressed vertically, and the labels are written with very small characters. Being small and compressed, the labels cannot be read clearly. The “exponential rise truncated at the insulating transition” mentioned at page 4 should rather be an exponential decay since the insulating transition occurs at low current values. In general, the description of Fig. 1c is not sufficiently detailed and/or clear despite it is a pivotal result of the present work.

We have made the Raman graph as big as possible given the space constraints. The phrase ‘exponential rise’ has been changed to ‘exponential decay’.

The Raman results (Fig. 1 and Supplementary Fig. 8) are not so clear and are not discussed in detail. Why are the spectra of the pristine sample (or de-hydrogenated/deuterated) characterized by a much lower signal to noise ratio? Furthermore, why does the G peak show a much lower intensity than the 2D peak (up to disappearing almost completely, as shown in Supplementary Fig. 8) in the pristine samples at variance with the hydrogenated/deuterated ones? And why is there such a huge variation in the peak width for both the G and 2D modes? In Supplementary Fig. 8, around 1300 cm^{-1} , there are some “negative” spikes whose intensity is comparable to that of the G peak. What are they due to? It looks like the signal is very low and the spectra were not integrated much. Why instead are the data in Fig. 3b much better (i.e., with a much higher signal to noise ratio)? Fig 3c is very small, but it seems that the width of the G peak is not as narrow as in the other figures and it is maybe consistent with the width of the G peak shown for the hydrogenated/deuterated samples in Figs. 1 and 2.

The Raman spectra of the devices are already discussed in refs. 14&15 and are used here only as a reference to confirm that the hydrogenation process is taking place. Nevertheless, following the Reviewer comment, we have expanded the discussion of these data in the manuscript.

The 2D band intensity is low because the concentration of protons is so high that graphene is now in the highly disordered/amorphous phase. In this regime, the ratio of integrated-area ratio of the D and G bands (A_D/A_G) is known to be greater than one (e.g. Luchese et al Carbon 2010, 48, 1592-1597 and Jorio et al, Physica Status Solidi B 2010, 247, 2980-2982). This is consistent with our findings, which reveal a peak intensity ratio of $I_D/I_G \approx 1.08$ and integrated-area ratio of $A_D/A_G \approx 2.03$, as shown in Fig. R1.1 (see response to point #6 from Reviewer #1, page 3). Note also that $I_D/I_G \approx 1$, $A_D/A_G \approx 2.03$ can translate both into a distance between H atoms of $L_D \approx 1\text{ nm}$ or $L_D \approx 10\text{-}15\text{ nm}$. This is a consequence of the bell-shape of the I_D/I_G vs L_D dependence (e.g. Fig. 4 in Luchese et al). To decide which one applies, it is necessary to look for additional signatures of disorder in the spectra. In our spectra, the 2D band is smeared, a signature of highly disordered systems, and thus consistent with $L_D \approx 1\text{ nm}$.

Regarding the question of why the G band intensity in pristine samples is much lower than for the 2D band, we point out that this is what is expected for pristine graphene. The ratio of 2D to G band is normally around 2-4 times (e.g. Physical review letters, 2006, 97: 187401; Physical Review B, 2010, 82: 125429).

Regarding the signal-to-noise ratio, ref. 14&15 already showed that the transition to highly disordered/amorphous graphene during the hydrogenation process is accompanied by an

increase in intensity of the overall Raman spectrum. This is very noticeable in the Raman data in Fig. 1 and Fig. S2 because the electrolyte yields a significant background. This background obscures the spectra of pristine graphene (this is the source of the ‘negative’ spikes in the spectra) but does not obscure the more intense signal from hydrogenated graphene. This background is not an issue for the data in Fig. 3c because those samples are not in contact with the electrolyte. This has been clarified in the corresponding figure captions in the manuscript. These latter spectra characterise the starting samples prior to hydrogenating them, to demonstrate that they are defect free. Following the Reviewer comment, we have included this discussion in Raman spectroscopy in Methods.

The right panel in Supplementary Fig. 8 shows a sudden change in the Raman spectrum, from the absence of the D band to its presence. What happens in between? Can the hydrogenation/deuteration process be monitored? And how precise is the information which one can get from the Raman spectrum? On page 4, it is said that a deuterium concentration of $\sim 10^{14} \text{ cm}^{-2}$ is deduced by the Raman spectrum. How precise is this concentration value? The hydrogenation process has already been monitored with Raman in this work and in refs. 14&15 and the transition was shown to be sudden – as expected from the sharp conductor-insulator transition. We now show that even fixing a relatively low potential of ≈ 1.5 vs NP would lead to hydrogenation of the sample within a couple of seconds, as shown in Fig. 1e. However, the dehydrogenation is a more gradual process, as already shown in ref. 14 and as we show in Fig. R1.1, in response to question 6 from Reviewer #1 (page 3).

Regarding the accuracy of the H (D) concentration in hydrogenated (deuterated) samples, these estimates are accurate within order of magnitude (e.g. Luchese et al Carbon 2010, 48, 1592-1597). However, the electronic transport measurements reveal that the hydrogenation transition takes place for a charge carrier density of $\approx 1 \times 10^{14} \text{ cm}^{-2}$ (the Fermi energy extracted from these measurements can be readily converted into charge density) in agreement with the Raman experiments. Hence, the concentration of H atoms in the hydrogenated phase can be estimated with a fair degree of accuracy.

Generally, information on the integration time and total intensity of the data should be provided (whenever normalized spectra are shown, it would be useful to say how large would the intensity factor be for hydrogenated/deuterated and pristine samples).

Not enough information on the Raman measurements is provided (laser wavelength, information on the setup and detectors, etc.).

We have specified the various parameters used in the Raman experiments in the methods section.

In Fig.2, it would be better to re-use for protons the same colors used in Fig. 1, e.g., red. This would also make the Figure clearer (the grey lines are not so well visible).

We have made this change.

The effects of the morphology on the electronic/electrochemical currents seems the most interesting result of this work. The way it is presented is though misleading. On one hand, the reader gets the impression that corrugated graphene is beneficial, but from our understanding it is instead detrimental because a larger electrochemical current is required. On the other hand, corrugated graphene shows a higher electronic current that seems to point to a beneficial effect. This part of the work should be better explained as to remove this seemingly inconsistency.

The electronic current in Fig. 3 is now normalised by the flake width in the revised manuscript. This reveals that the electronic current is the same for all three types of devices within experimental scatter (see point 4 from Reviewer #1) and thus that corrugations do not result in higher electronic current.

Corrugations do have a significant effect on the electrochemical hydrogenation process. The electrochemical current of the devices is strongly enhanced, and the onset of the process is about -1 V vs NP for corrugated samples, compared to about -1.5 V vs NP for flat graphene. Our DFT calculations show that this arises because convex areas within the corrugations have a lower energy barrier for proton adsorption, which leads to a large reduction current via an Eley-Rideal process. This reduction process stops when the insulating transition takes place, which happens at potentials comparable to those observed in flat graphene (graphene-on-hbn). Our DFT calculations suggest that this arises because corrugated samples contain both convex and concave areas – and the latter do not favour hydrogenation. This results in the preferential accumulation of protons in convex areas, leaving concave ones free from hydrogen atoms. This delays the overall insulating transition, which requires all electrically conductive pathways in the flake to be closed.

In summary, corrugations locally enhance the proton adsorption rate, but the conductor-insulator transition of the whole flake is dominated by the concave areas in the flake, whose hydrogenation becomes the limiting step in the transition.

In conclusion, the manuscript requires important revisions, and, in particular, it is not of adequate general interest for Nature Communications, but it is more suitable for a specialised journal.

We have done our best to address all the points raised by the Reviewer, which we believe have improved the clarity of our manuscript. We would also like to emphasise the relevance of our work, which addresses the fundamental question of how protons adsorb to graphene electrochemically and which, as pointed out by Reviewer #1, is of great relevance and interest. We hope that in light of these revisions and extended discussions, the Reviewer might consider recommending our manuscript for publication.

Response to comments from Reviewer #4

I co-reviewed this manuscript with one of the reviewers who provided the listed reports. This is part of the Nature Communications initiative to facilitate training in peer review and to provide appropriate recognition for Early Career Researchers who co-review manuscript

We thank the Reviewer for taking the time to revise our manuscript.

Response to comments from Reviewer #1

We thank the authors for their responses. We still have several questions after the authors have revised their manuscript.

We thank the Reviewer for taking the time to do a careful review of our manuscript. We have made our best to answer all questions raised.

1) Thanks to the authors for redrawing the circuit, which makes it clearer. The authors used a source meter (Keithley 2636B source meter, two-wire measurement) and a voltmeter (Keithley 2182A Nanovoltmeter) instead of using a potentiostat to perform CV scans. We believe that this platform, particularly the reference electrode, cannot provide feedback control for the current loop. The circuits in the references 1,2 seem to also use a source meter with a voltmeter. We therefore still question whether the use of a potentiostat would yield different results than using two separate Keithley instruments for determining hydrogenation overpotentials. In electrochemistry we use potentiostat as standard equipment.

Our setup can be programmed to function as a potentiostat, as mentioned in the Methods section. More information can be found on the manufacturer's manual [https://www.tek.com/en/documents/application-note/keithley-instrumentation-](https://www.tek.com/en/documents/application-note/keithley-instrumentation-electrochemical-test-methods-and-applications)

[electrochemical-test-methods-and-applications](https://www.tek.com/en/documents/application-note/keithley-instrumentation-electrochemical-test-methods-and-applications).

To confirm this capability, Fig. R1.1 shows measurements of the classic $K_3Fe(CN)_6$ redox reaction vs Ag/AgCl reference electrode with our setup. The advantage of our setup is its high adaptability, which enables performing both electronic and electrochemical measurements simultaneously.

Fig. R1.1. Cyclic voltammogram of Pt wire working electrode immersed in a 10 mM $K_3Fe(CN)_6$ aqueous solution with 1.0 M KCl supporting electrolyte. Reference electrode Ag/AgCl (3M NaCl). Scan rate 20mV/s. The figure shows a separation between reduction and oxidation peaks (ΔE_p) of 77 mV, as expected.

2) In their response to point 2 the authors mention that the conductivity of deuterons in aqueous electrolytes is ~10% smaller than for protons (Arcis et). According to the CRC Handbook of Electrochemistry and Physics 3 (page 5-77), the conductivity of H^+ is 1.4 times that of D^+ in water at 25°C.

We are aware of the 1.4 factor. However, this value applies to aqueous electrolytes and may differ for HTFSI and DTFSI in PEG. In our response we used the symbol “~” and the 10% value to reflect this uncertainty. The important point is that the difference in conductivity between H^+ and D^+ is expected to be relatively small.

3) Separately, in Figure S6, the authors could not measure a significant difference of conductivity between H^+ and D^+ in HTFSI and DTFSI for SiN_x substrates with a hole without graphene, and attributed this result to device-to-device variations (see point 2, where a difference of 1.4 is reported for bulk conductivities of H^+ and D^+). Therefore, we question whether PdHx and PdDx are reliable in detecting proton or deuteron currents.

PdHx/PdDx electrodes have been shown to be effective to measure proton/deuteron currents (e.g. C. Zhong, et al. Nat. Comms. 2011, 2:476). However, we completely agree with the Reviewer that the contact resistance of these electrodes will depend on hydrogen loading. This could indeed contribute to the ~10% error observed in our open-hole measurements (without graphene).

However, our data indicate that if contact resistance variations exist, they have a negligible impact on experiments with graphene. This is evident from the low device-to-device variability in both the potential of the hydrogenation transition and the peak currents in the electrochemical signal.

This robustness was expected. In our experiments, the working electrode (graphene) is highly resistive, with currents orders of magnitude lower than in open-hole devices. Additionally, the graphene electrode is micrometer-sized, while the counter electrode (PdHx) is centimeter-sized. In this regime, cell resistance is dominated by the microelectrode (e.g., ref. 40 in the text). As a result, any small variations in contact resistance due to different PdHx electrodes should have minimal impact on device performance, as confirmed by our observations.

4) In our initial question we asked which electrochemical reactions occur on PdHx. We asked this question because we believe that the electrochemical reactions occurring at the reference electrode are key to determine the overpotential of hydrogenation on graphene. We asked the following question: “what are the specific reaction equations that occur on PdHx when gaining and losing electrons?” In their revised manuscript, the authors mention that “Pd + H++ e- -> PdHx” is happening. What if the reaction would be “Pd + xH+ + xe- -> PdHx. Separately, if PdHx is saturated with H, atomic hydrogen or hydrides can no longer be inserted into the electrode. The literature report on an upper limit to the electrochemical insertion of H into Pd, which could be considered here for the discussion.⁵ Possible reactions include 6:(see pdf)

$$2 \text{ H}^{++} + 2 \text{ e}^{-} \rightarrow \text{H}_2 \quad (1)$$

$$\text{[PdH]}_x + \text{H}^{++} + \text{e}^{-} \rightarrow \text{Pd} + \text{H}_2 \quad (2)$$

$$\text{[PdH]}_x \rightarrow \text{Pd} + \text{H}^{++} + \text{e}^{-} \quad (3)$$

$$\text{H}_2 \rightarrow 2 \text{ H}^{++} + 2 \text{ e}^{-} \quad (4)$$

Unfortunately, we cannot determine definitively which of these reactions is taking place, and any characterisation we could perform is unlikely to resolve the question conclusively. However, we point out that these suggested reactions primarily take place in hydrogen-saturated PdHx electrodes or nanoscale Pd, which is likely not the case here. More important, as noted in the references suggested by the Reviewer and several others (e.g., ref. 6 from the Reviewer; C. Gabrielli et al., *J. Electrochem. Soc.* 2004, 151, A1937; K. Hubkowska et al., *Electrochim. Acta* 2010, 56, 235-242), the onset potential for these reactions typically differs by 100–200 mV. This would result in a one-off shift of the same magnitude in the potential of our devices versus SHE, which is acceptable for the purposes of this work. Note as well that the difference in potential for the insulating transition triggered with proton- and deuteron-conducting electrolytes is 300 mV and thus larger than the uncertainty in question.

Independent of all this, a key advantage of measuring both the electronic and electrochemical responses of our devices simultaneously is the ability to reference the potential of the electrochemical processes against the NP in graphene. This provides access to the absolute scale, which is independent of the reference electrode and remains constant across all devices and processes because it depends solely on the band structure of graphene. This method thus provides a direct way to characterize these processes that bypasses the uncertainties that may be associated with non-aqueous reference electrodes. We believe that this is an important novel

aspect of our work, which could be applied to other processes. Following the Reviewer's comment, we have stressed the importance of the electronic response of graphene as an independent reference in page 2, second paragraph, line two before last.

5) Overall and over the time of the measurement, the composition of the reference electrode may change and therefore such reference electrode may not be used to determine the overpotential of hydrogenation on graphene. In order to determine whether the reference electrode composition changes (or not), the authors could use XRD or XPS before and after the measurement. The PdH_x electrodes are stable over time in our experiments. This is evidenced from the fact that the position of graphene's neutrality point does not change over time in our measurements. It is also evident from the low variability of the hydrogenation potential between different samples. As discussed in point 3, this stability is expected from the huge difference in size between the working and counter electrode.

6) In the lower panel of figure 3C, corrugated graphene shows a higher current compared to graphene on hBN and graphene on SiO₂; the authors carried out an error analysis and concluded that differences in currents are attributed to device-to-device variations (now the author normalized the electronic current to the width of the graphene). We expect corrugated graphene to have lower charge carrier mobilities than less corrugated graphene. What if the authors would calculate and compare the carrier mobility for the different samples instead of the current 7,8 ; and establish whether corrugated graphene has a better affinity for H⁺ (see also point 5). Our work studies the conductor-insulator transition induced by hydrogen adsorption. The mobility of electrons is unnecessary to understand this phenomenon. Nevertheless, regarding the question of whether corrugations are expected to change graphene's mobility, we note that at low temperatures, one indeed expects corrugated samples to have lower mobility. However, at ambient conditions this is not expected. Theoretically, mobility at these temperatures is primarily limited by phonon scattering (in the absence of defects), and corrugations are not expected to alter the already strong scattering at this temperature. For a broader discussion on this subject, see S. Das Sarma et al., *Rev. Mod. Phys.* 2011, 83, 407-470. For an explicit calculation of this effect at room temperature see, for example, in R. Shah & M. G. Mohiuddin, *Proc. World Cong. Eng.* 2011, Vol II (https://www.iaeng.org/publication/WCE2011/WCE2011_pp1357-1362.pdf). Additionally, we point out that extracting carrier mobility – especially subtle effects that could arise from lattice corrugations – requires Hall effect measurements. Estimates based on two probe resistance measurements cannot provide conclusive insights into this question.

7) Corrugated graphene should have much lower overpotential of hydrogenation compared to graphene on SiO₂ and hBN. However, corrugated graphene was measured to have the similar overpotential as for graphene on SiO₂ and hBN (the potential at the peak of hydrogenation current in upper panel of Figure 3c). Similarly, corrugated graphene should have much lower overpotential at the insulating transition compared to graphene on SiO₂ and hBN. However, corrugated graphene was measured to have the maximum overpotential (the potential at the zero current transition in lower panel of Figure 3c) compared to graphene on SiO₂ and hBN. The authors explain these results as "concave areas relatively free of protons [...] leaves electronically conductive paths". However, this reason is unclear. Drawing a schematic diagram of electronically conductive paths in the concave and convex areas may be relevant. In addition, if the concave areas with conductive paths exist, why does corrugated graphene undergo the same insulating transition as graphene on SiO₂ and hBN?

To clarify this observation, it is important to note that the electrochemical process driving the hydrogenation (insulating) transition is proton adsorption, which competes with desorption via an Eley-Rideal process. The insulating transition occurs when, as a result of this process, a high proton coverage of $\sim 10^{14} \text{ cm}^{-2}$ is reached.

As the Reviewer correctly expects, corrugated graphene indeed exhibits a significantly lower onset potential for the electrochemical process. To illustrate this more clearly, we provide the electrochemical response in Fig. 3, zoomed in on the current range where the reaction begins (Fig. R1.2). This figure shows that the reaction onset is indeed lowest for corrugated graphene, followed by graphene-on-SiO₂, and finally graphene-on-hBN.

Fig. R1.2. Electrochemical response of different graphene samples zoomed in from Fig. 3 at the current level where the proton adsorption onset.

The unexpected finding is that, while the onset for proton adsorption is lower in corrugated graphene compared to graphene-on-hBN or graphene-on-SiO₂, all samples require similar potentials to achieve coverage of $\sim 10^{14} \text{ cm}^{-2}$. We attribute this to the

existence of areas in corrugated graphene that resist hydrogenation, thus preventing the sample as a whole to achieve proton coverage of $\sim 10^{14} \text{ cm}^{-2}$. Our DFT calculations suggest that such areas are probably those with concave corrugations, where proton adsorption requires similar potentials to those of flat graphene (Supplementary Fig. 1). This would explain why full coverage is achieved only at potentials comparable to those observed in graphene-on-hBN or graphene-on-SiO₂.

While we could provide schematics of this interpretation, we believe they would not add significant clarity. A definitive answer to this question would require techniques capable of measuring proton coverage with high spatial resolution (e.g., STM). However, this is beyond the scope of this work. In response to the Reviewer's comment, we have edited the discussion in the main text of the manuscript, last paragraph before conclusion (pages 6&7), to clarify these points. We have also added Fig. R1.2 as inset in Fig. 3.

8) The size of the washer compared to the size of graphene is very important for the fully insulation transition. So, we asked before, "Are the graphene samples studied here bigger or smaller than the opening in the SU8 washer?" The authors point out that the graphene flake must be smaller than the opening of the SU8 washer, which is important for the insulating transition. We strongly agree with this point of view. However, in a previous paper by the same research group 1, the graphene flake seems larger than the opening in the SU8 washer, and an insulating transition is also observed. Are there here any contradictions between the two situations/articles? The washers in both this work and ref. 1 are either larger or of the same size of the flake. The device shown in Extended Data Fig. 1 of ref. 1 was a device in which the size of the flake matched that of the washer. There are no contradictions between the two works.

References

1. Tong, J. et al. Control of proton transport and hydrogenation in double-gated graphene. *Nature* 630, 619–624 (2024).
2. Cai, J. et al. Wien effect in interfacial water dissociation through proton-permeable graphene electrodes. *Nat Commun* 13, 5776 (2022).

3. CRC Handbook of Chemistry and Physics. (CRC Press, 2014). doi:10.1201/b17118.
4. Lozada-Hidalgo, M. et al. Sieving hydrogen isotopes through two-dimensional crystals. *Science* (1979) 351, 68–70 (2016).
5. Benck, J. D., Jackson, A., Young, D., Rettenwander, D. & Chiang, Y.-M. Producing High Concentrations of Hydrogen in Palladium via Electrochemical Insertion from Aqueous and Solid Electrolytes. *Chemistry of Materials* 31, 4234–4245 (2019).
6. Moumaneix, L., Rautakorpi, A. & Kallio, T. Controlling the Reactivity and Interactions between Hydrogen and Palladium Nanoparticles via Management of the Particle Diameter. *ChemElectroChem* 10, (2023).
7. Zhang, Z., Xu, H., Zhong, H. & Peng, L.-M. Direct extraction of carrier mobility in graphene field-effect transistor using current-voltage and capacitance-voltage measurements. *Appl Phys Lett* 101, (2012).
8. Zhong, H., Zhang, Z., Xu, H., Qiu, C. & Peng, L.-M. Comparison of mobility extraction methods based on field-effect measurements for graphene. *AIP Adv* 5, 057136 (2015).

Response to comments from Reviewer #2

We thank the Reviewer for taking the time to review our manuscript.

Response to comments from Reviewer #3

The authors replied to the previous concerns and the manuscript has improved, yet some aspects regarding Raman measurements remain not fully clarified, such as the quality of the spectra. In particular, while it has now been clarified that the spectra of the pristine samples are affected by the presence of a electrolyte background signal, this should not impede to have a better signal-to-noise ratio. In the methods section, it is mentioned that for Raman measurements, "accumulating time of 75s over 4 cycles" were used. Were both the pristine and hydrogenated sample spectra taken under these same conditions?

We thank the Reviewer for taking the time to review our manuscript. We believe the comments have improved it.

Regarding the question, both the pristine and hydrogenated sample spectra were measured with the same conditions. Unfortunately, we did not manage to enhance the resolution of the spectra for pristine graphene in the presence of the electrolyte. Changing different parameters such as laser power (0.4 mW to 1 mW), accumulating time (10-100 s) and cycles (1-5), did not make a notable difference. However, the key spectral feature that characterises the process is the I_D/I_G ratio. This feature is clearly discernible in the acquired spectra, rendering further enhancement unnecessary for the purposes of this work.

In the caption of figure 1, it is said "Raman spectra of pristine and hydrogenated samples. The background signal from the electrolyte was subtracted; the latter spectrum was divided by 5 for clarity". From this, it seems that there is only a factor of 5 in the intensity of the spectra, while the signal-to-noise-ratio seems to be lower than that by looking at the spectra. Furthermore, the spectra in Fig. 3b look better than those in Fig. 1 and Supplementary Fig. 8. Is this discrepancy due to different accumulation times or to something else?

To clarify this point, Fig. R2.1 shows the Raman spectra for the different samples taken under the same conditions, without normalisation factor (factor of 5). It shows why dividing the spectra of hydrogenated graphene by a factor of 5 is helpful for presentation purposes.

Fig. R2.1. Raman spectra of hydrogenated graphene (blue data) and pristine graphene (red) in contact with the electrolyte. No normalisation constant was applied to the data. For reference, we show the spectra of pristine graphene prior to exposing it to the electrolyte (black).

Regarding the lower noise of the Raman data in Fig. 3b compared to Fig. 1, we point out that, as mentioned in our previous response, these samples were not exposed to the electrolyte. As for the data presented in Supplementary Fig. 8, its signal-to-noise ratio is comparable to that of Fig. 1. It may appear clearer because it is displayed in a larger

panel. However, as previously noted, space constraints in the inset of Fig. 1 prevent displaying the figure any larger.

As for Figure 1, we know that the data of Figure 1 are different to those shown in 2024 Nature paper. In our previous report, we were referring to Fig. 1a (now Fig. 1d) showing the Energy vs distance plot. This plot shows the same data plotted in Extended Data Fig. 8 (zero electric field). We are sorry we were not clear enough.

Thank you for clarifying this point. The energy vs. distance plot in Fig. 1d is a classical calculation that appears in numerous papers, including refs. 11-15 and Extended Data Fig. 8 in Tong et al. However, we believe it is important to include this plot in the main text of our work, as it plays an important role in the discussion. This is especially relevant given that our manuscript is intended for a broad audience, which may not be familiar with the hydrogenation of graphene.

This said, while we believe that the manuscript was improved, we still feel that this work would be more suitable for a specialised journal.

We thank the Reviewer for their comments, which we believe have improved our manuscript. Regarding the suitability of our work for this journal, we respectfully disagree, but we will not revisit this point further, as this was addressed in the previous response.

Response to comments from Reviewer #4

We thank the Reviewer for taking the time to review our manuscript.

Response to comments of Reviewer #1

Below are my detailed comments and suggestions, which may be helpful for revising the manuscript:

We thank the Reviewer for their careful evaluation of our manuscript, as well as for their continued support for our work throughout the revision process. In response to the latest comments, we have added several new experiments and expanded key discussions. We believe these additions address all the points raised and further strengthen the clarity of the manuscript.

1. Circuit reproducibility and experimental setup

We thank the authors for elaborating on their circuit and introducing a discussion on using the Keithley 2636B as a potentiostat. However, the current explanation remains ambiguous. It would significantly improve clarity and reproducibility if the authors could provide a schematic diagram showing how the Keithley 2636B sourcemeter and the 2182A nanovoltmeter are connected to the graphene and PdHx electrodes. This would assist future researchers in reproducing the setup accurately. Still, I would not be surprised that doing similar experiments with a potentiostat would yield different onset potentials for hydrogenations and dehydrogenations and therefore results/interpretations/conclusions.

Additionally, could the authors clarify whether the Keithley 2636B sourcemeter was operated in 2-wire or 4-wire mode? As the manufacturer's manual notes, the sourcemeter functions as a potentiostat only in 4-wire mode, where the Sense LO provides feedback to Force LO. If used in 2-wire mode, this feedback loop is not valid, which would compromise the reliability of the measured potentials at the reference electrode.

see Figure 2 of this manual on the website of Tek:

Fig. R1. a, Sample-to-Keithley connection diagram. **b**, Electrochemical current-voltage response of a sample measured with the Keithley setup and an Ivium pocketSTAT2 potentiostat.

The Keithley was operated in 4-wire mode for all measurements. We have added a detailed wiring diagram in the revised manuscript (Figure S2). The mention of sourcemeter and nanovoltmeter was an oversight, carried over from (Tong, J et al., *Nature*. 2024, 630, 619-624). In that study the electrolytic current involved proton transport without electron transfer, so no potentiostatic

control was needed—in any case, gate/reference potentials differed by less than 4 mV in that study (Extended Data Fig. 2 in Tong, J et al., *Nature*. 2024, 630, 619-624).

To ensure full clarity, and to directly address the Reviewer's concern, we have now also performed cyclic voltammetry measurements of graphene hydrogenation using an Ivium pocketSTAT2 potentiostat. As shown in Fig. R1, these measurements reproduce the same redox features and potentials observed in our previous data, confirming the consistency and reliability of our approach. The only minor difference is that the Keithley setup offers better resolution at the lowest current levels (~1 pA), which are near the detection limit of the Ivium system. We thank the Reviewer for drawing our attention to this point. We hope this work will help popularising the use of sourcemeters as highly versatile instruments for electrochemical measurements.

2. Use a potentiostat to perform CV on graphene hydrogenation

While we appreciate the CV measurements provided for the Pt electrode in a $K_3Fe(CN)_6$ solution (Fig R1.1 of the rebuttal), this alone, without a counter electrode does not validate the electrochemical circuit used in the manuscript. If the authors aim to demonstrate the validity of their custom circuit, they could consider performing measurements using a commercial potentiostat on graphene hydrogenation with PdHx electrodes and compare the results obtained with their setup using the two Keithleys.

In the CV measurements with $K_3Fe(CN)_6$ in the previous round of comments, the counter electrode was a graphite rod, validating the electrochemical circuit. However, we now understand the insistence on this point. Following the reviewer comment, we measured the hydrogenation of graphene using an Ivium potentiostat as described above.

Taken together, our results confirm that our setup reproduces both classical electrochemical experiments and the new hydrogenation reaction in graphene. We trust that, together with the schematics provided, these new experiments resolve the reviewer's concern in full.

3. Ambiguity in proton and deuteron transport

The manuscript should clarify the differences observed between H⁺ and D⁺ transport in free pore measurements in the context of mobility similarities/differences in HTFSI/PEG systems or whether those differences may arise from the electrode–electrolyte contact resistance. A more detailed discussion or additional control experiments would help clarify the interpretation of the data.

Figure R2. Conductance of devices with circular apertures etched in a silicon nitride substrate; HTFSI/DTFSI electrolyte on both sides; and two PdHx/PdDx electrodes. Each data point is a different device. Solid lines, best linear fit to data.

We have performed additional measurements using SiNx substrates with holes of different radii, r . This allows us to extract the electrolyte conductivity, κ , from the measured conductance, G , using the equation: $G = 4\pi\kappa r$ (Bard, A. J. & Faulkner, L. R., *Wiley*. 2001). This reveals $\kappa \approx 1.062 \times 10^{-4} \text{ S cm}^{-1}$ for HTFSI, consistent with other studies (e.g. Yang, Z., et al., *J. Membr. Sci.* 2008, 313, 91-96; Yu, L., et al., *J. Phys.*

Chem. B. 2012, 116, 6553-6560; Yoshizawa, M. & H. Ohno., *Chem. Commun.* 2004, 1828-1829). More importantly, with this expanded dataset, we are now able to resolve a ~35% lower conductivity in the DTFSI electrolyte.

We have been unable to find any studies on the isotope effect in H^+/D^+ in this electrolyte's conductivity and as such, we believe this data set is an additional novelty ingredient of our work. We have added this discussion in the Methods section, page 7 and Fig. R2 is the new Fig. S6.

4. PdHx electrode characterization and proton transport in HTFSI/PEG

The PdHx electrode is used for detecting hydronium ion currents, as demonstrated in prior literature involving hydrated media (e.g., maleic–chitosan, ref 1). However, PEG is not hydrated. There must be some difference between hydronium ion and HTFSI when those interact with PdHx. The manuscript should carefully discuss how proton exits in HTFSI and how they react with PdHx electrodes.

Following the Reviewer comment, we have added a new section to the manuscript in which we discuss proton transport in HTFSI as well as proton interaction with PdHx electrodes. Below, we summarise the key points.

Proton transport in PEG/HTFSI. Protons originate from HTFSI, a superacid ($pK_a \approx -10$), and exist as either free H^+ or PEG-coordinated complexes. Proton transport is thought to take place primarily via a Grotthuss-type mechanism enabled by transient hydrogen bonding with its ether oxygens (Moses, A. A. & Arntsen, C., *Phys. Chem. Chem. Phys.* 2023, 25, 2142-2152; Burankova, T. et al., *J. Phys. Chem. B.* 2015 119, 10643-10651). This is consistent with our conductivity measurements with open hole devices that demonstrate a notable isotope effect (see point #3 above).

Proton interaction with PdHx in aqueous electrolytes. Some of the first investigations of palladium (Pd) and hydrogen interactions were the classical experiments by Flanagan (Flanagan, T. B. & Lewis, F. A., *J. Chem. Phys.* 1958, 29, 1417-1418), which demonstrated hydrogen absorption in Pd films exposed to pressurised H_2 gas. Under these conditions, Pd forms two distinct hydride phases: the low-hydrogen content α -phase and the hydrogen-rich β -phase. In bulk Pd, these phases often coexist, and their distribution depends on hydrogen pressure, temperature, and the electrode's morphology, as described by the material's phase diagram (Flanagan, T. B. & Lewis, F. A., *J. Chem. Phys.* 1958, 29, 1417-1418).

Fig. R3. Potential of PdHx electrode vs SHE in 1M HCl in H_2 atmosphere as a function of time. Potential at zero current is recorded in a three-electrode cell with PdHx working electrode and Pd counter electrode and Ag/AgCl (3M) reference electrode.

Flanagan also examined the equilibrium between PdHx and acidic solutions in H_2 environment. Pd electrodes saturated with hydrogen exhibit an open-circuit potential close to 0 V vs SHE—reaching exactly 0 V when fully saturated. This reflects equilibrium between surface-adsorbed hydrogen and protons in solution via the Volmer step: $H^+ + e^- + * \rightarrow H^*$. Hence, hydrogen-saturated Pd electrodes behave like Pt in acidic solutions. Conversely, pristine Pd initially shows a potential of ≈ 90 mV vs SHE, which gradually decreases to 0 V over tens of hours as the electrode becomes electrochemically hydrogenated (Flanagan, T. B. & Lewis, F. A., *J. Chem. Phys.* 1958, 29, 1417-1418). Electrodes with intermediate hydrogen content exhibit potentials between 0 and 90 mV vs SHE, indicating coexistence of α and β phases. We characterised our electrodes in this way. Fig.

R3 shows that our electrodes have a potential of ≈ 61 mV vs SHE, which would correspond to intermediate H loading.

Proton interaction with PdHx in non-aqueous solvents. In non-aqueous solvents, the same equilibrium principles apply at the Pd/electrolyte interface. Hubkowska, K., et al., *J. Solid State Electrochem.* 2024, 28, 1159-1169 shows that these reactions proceed more slowly than in aqueous media and the IV characteristics of the electrode can shift slightly compared to aqueous electrolytes. Thus, the underlying redox mechanism at PdHx remains the same, except with slower kinetics and slightly different potentials. Our devices bypass the minor uncertainties associated with reference electrodes in non-aqueous by offering direct access to the absolute potential scale via the graphene neutrality point. As demonstrated by the statistical analysis in Table S1, the hydrogenation potential vs the neutrality point is highly reproducible and is in agreement with the previous independent study of this process (Li, S. et al. *Nat Electron.*, 2021, 4, 254-260). This discussion has been added in a new section, ‘Proton transport in the HTSFI/PdHx system’ in Methods, page 9.

Moreover, it is unclear how the authors prepared the PdHx electrode. In previous studies, the PdHx electrode were saturated with hydrogen (ref 1–3). If saturation occurred, the ability of PdHx to absorb further protons would be limited. Could the author describe more in details how they prepared the PdHx electrodes.

We prepared PdHx following the protocol described and used in Murphy, D. W. et al. *Chem. Mater.*, 1993, 5, 767-769. In both works, palladium is immersed in a sodium borohydride solution overnight while stirring and thoroughly rinsed with deionised water. We have added this reference to the manuscript in the Methods section.

Additionally, the electrochemical equation on page 7 appears unbalanced. A more rigorous form would be: $xH^+ + x e^- + Pd \rightarrow PdH_x$. This should be corrected for clarity and chemical accuracy.

We agree with the Reviewer and have written the reaction as suggested.

5. Mobility estimation and interpretation of insulating behavior

The authors do not use two-end (source and drain) field-effect-based mobility analysis, despite its widespread adoption in similar studies (Ref 4–8). If there is a known inconsistency in applying this method to the present system, it should be discussed explicitly. Avoiding this issue weakens the interpretation of the observed insulating transition, which is based on carrier conduction but lacks support of quantitative carrier mobility and carrier concentration.

Following the Reviewer’s suggestion, we have estimated the electronic mobility in our samples using the model in Lv, H. et al. *Appl. Phys. Lett.*, 2013, 103, 193102. The carrier concentration in graphene, n , is:

$$n = \sqrt{n_0^2 + n(V_G^*)^2} \quad (1)$$

where n_0 and $n(V_G^*)$ is the induced carrier concentration as a function of the applied gate voltage, V_G vs the neutrality point. The relation between gate voltage and induced carriers is:

$$V_G^* = \frac{e}{C_E} \cdot n(V_G^*) + \frac{\hbar \cdot v_F \cdot \sqrt{\pi n(V_G^*)}}{e} \quad (2)$$

where e and \hbar are the elementary charge and reduced Planck constant, respectively. v_F is the Fermi velocity, of 1.15×10^6 m s⁻¹ and $C_E = 20$ μ F cm⁻² is the electrolyte capacitance, as shown in our previous work.

The total resistance, R_{Total} , is then written as:

$$R_{Total} = R_C + \frac{L_G}{W_G} \cdot \frac{1}{e \cdot \mu \cdot n} \quad (3)$$

where R_C is the contact resistance and L_G and W_G refer to the gated channel length and width, respectively and μ is the mobility to be extracted.

Fig. R4. Total resistance as a function of the gate voltage vs NP for graphene on SiO₂ (left) and corrugated graphene (right).

We have estimated carrier mobilities in our samples with this model, as listed in Table R1 below. The extracted mobilities are in the range of 2,000 - 6,000 cm²/V·s, in good agreement with the literature (K. S. Novoselov et al., Science, 2004, 306,666-669) for similar samples (~1,000 to 10,000 cm²/V·s). The estimates also suggest a small ~2× lower carrier mobility for corrugated graphene compared to graphene-on-SiO₂ devices. This information has been included in the revised manuscript, Methods section, page 11.

Table R1. Estimated carrier mobility in graphene samples

	Electron mobility 10 ³ cm ² / V·s	Hole mobility 10 ³ cm ² / V·s	Contact resistance kΩ	Residue carrier concentration 10 ¹¹ cm ⁻²	Number of samples
Graphene-on-SiO ₂	3.800 ± 0.8	5.9 ± 0.7	2.2 ± 0.3	8.0 ± 1.1	8
Corrugated graphene	1.9 ± 0.5	2.8 ± 0.5	2.0 ± 0.3	6.9 ± 1.4	4

6. Role of curvature in hydrogenation

The authors emphasize that curvature (convex and concave regions) affects graphene hydrogenation, yet they don't provide direct visual or experimental support for this claim. Including schematics or Raman mapping (e.g., spatial or time/potential-resolved analysis of D-band changes) could help clarify whether such morphological effects exist and how they influence reactivity.

Morphological effects on the hydrogenation of graphene have previously been reported in studies using plasma as the hydrogen source. For example, Goler, S. et al., *J. Phys. Chem. C.* 2013, 117, 11506-11513 performed STM measurements on plasma-hydrogenated graphene with local corrugations induced by the underlying SiC substrate. The authors visually demonstrated that hydrogen adsorption occurs preferentially in convex regions of the sample (e.g., Fig. R5). Away from these convex regions no hydrogen adsorption was detected. These findings are supported by DFT calculations, including those we present in Fig. S1, which predict a lower energy barrier for chemisorption on convex areas of the sample.

[Figure Redacted]

Fig. R5. STM measurements of plasma hydrogenated graphene on SiC substrates obtained from Goler, S. et al., *J. Phys. Chem. C.* 2013, 117, 11506-11513. Scale bar, 2 nm.

Our results are consistent with these observations. To further support this point, we have performed potential-resolved Raman spectroscopy measurements that directly monitor the hydrogenation process, as suggested by the Reviewer. These spectra show a clear onset of the D band at the potentials corresponding to the hydrogenation of the sample. These new data are included as Fig. S10 in the revised manuscript.

Fig. R6. Raman spectra of corrugated graphene samples during a hydrogenation cycle. The signatures of the hydrogenation appear at about -2.1 V vs NP.

7. Novelty

Fig 3c shows a largely enhanced dehydrogenation current for corrugated graphene. This is the impressive part of the results. However, mechanistically, the hydrogenation of graphene has been reported in multiple prior studies (e.g., Refs.9–15). Several mechanistic discussions are also available in literature (e.g., Refs.16,17). It is still not clear from the manuscript and data what is the mechanism of hydrogenation and dehydrogenation.

Would the author measure similar high electrochemical currents in dry Nafion/PdHx systems (ref3)? Again I am not sure the electrochemistry is right in HTFSI (see above). Here also, I wonder whether switching to using a potentiostat would yield different results? However, I acknowledge that from Fig 3c top panel, the results are impressive.

We thank the Reviewer for their encouraging assessment of our results. This work builds on over a decade of investigations into proton-graphene interactions across a range of device configurations.

Fig. R7. Electrochemical and electronic response of graphene device gated with Nafion under dry Ar environment. Insets show that the devices do not survive the first hydrogenation cycle.

Regarding the mechanism, we agree that hydrogenation of graphene via plasma is well established, as discussed in Refs. 9–15 from the Reviewer and the introduction of our manuscript. In contrast, the electrochemical hydrogenation of graphene is a much more recent development, and its mechanism is not established. While Ref. 16 from the Reviewer (Daniels et al) and other studies report electrochemical and Raman measurements of graphene in aqueous environments, we note that they did not observe a conductor-insulator transition or demonstrate the reversibility in the Raman D band—both of which are critical hallmarks of successful

hydrogenation. In the absence of this evidence, it is premature to conclude that electrochemical

hydrogenation was achieved in those works. As such the mechanism for this phenomenon remains unexplored.

Following the Reviewer's suggestion, we have now performed the experiments using Nafion in dry Ar environment. Fig. R7. shows that the conductivity of the devices starts decreasing at around 1.8 V vs NP. Unfortunately, the devices break before reaching the insulating transition (insets in Fig. R7). This arises because even in dry Nafion films, water remains essential for proton conductivity (Mauritz, K. A. & Moore, R. B., *Chem. Rev.* 2004, 104, 4535-4586), and the large applied potential results in water breakdown.

The drop in conductance could perhaps be attributed to the onset of the hydrogenation process. The absence of an insulating transition may arise because water-based electrolytes, being more active (see point #4), also yield a faster Eley-Rideal process. However, the drop in conductance could also arise due to irreversible damage of the graphene electrodes. Since the devices do not display a full hydrogenation-dehydrogenation cycle, it is not possible to conclude based on these data.

1. Zhong, C. et al. A polysaccharide bioprotonic field-effect transistor. *Nat Commun* 2, 476 (2011).
2. Morgan, H., Pethig, R. & Stevens, G. T. A proton-injecting technique for the measurement of hydration-dependent protonic conductivity. *J Phys E* 19, 80–82 (1986).
3. Hu, S. et al. Proton transport through one-atom-thick crystals. *Nature* 516, 227–230 (2014).
4. Lv, H. et al. High carrier mobility in suspended-channel graphene field effect transistors. *Appl Phys Lett* 103, (2013).
5. Xu, H. et al. Top-Gated Graphene Field-Effect Transistors with High Normalized Transconductance and Designable Dirac Point Voltage. *ACS Nano* 5, 5031–5037 (2011).
6. Kim, S. et al. Realization of a high mobility dual-gated graphene field-effect transistor with Al₂O₃ dielectric. *Appl Phys Lett* 94, (2009).
7. Farmer, D. B. et al. Utilization of a Buffered Dielectric to Achieve High Field-Effect Carrier Mobility in Graphene Transistors. *Nano Lett* 9, 4474–4478 (2009).
8. Liao, L. et al. High- κ oxide nanoribbons as gate dielectrics for high mobility top-gated graphene transistors. *Proceedings of the National Academy of Sciences* 107, 6711–6715 (2010).
9. Elias, D. C. et al. Control of Graphene's Properties by Reversible Hydrogenation: Evidence for Graphane. *Science* (1979) 323, 610–613 (2009).
10. Li, S. et al. Large transport gap modulation in graphene via electric-field-controlled reversible hydrogenation. *Nat Electron* 4, 254–260 (2021).
11. Tong, J. et al. Control of proton transport and hydrogenation in double-gated graphene. *Nature* 630, 619–624 (2024).
12. Zhao, M., Guo, X.-Y., Ambacher, O., Nebel, C. E. & Hoffmann, R. Electrochemical generation of hydrogenated graphene flakes. *Carbon N Y* 83, 128–135 (2015).
13. Fei, Y., Fang, S. & Hu, Y. H. Synthesis, properties and potential applications of hydrogenated graphene. *Chemical Engineering Journal* 397, 125408 (2020).
14. Hu, Y. et al. Synaptic transistor based on reversible hydrogenation of graphene channel. *Appl Phys Lett* 126, (2025).
15. Cha, J., Choi, H. & Hong, J. Damage-free hydrogenation of graphene via ion energy control in plasma. *Applied Physics Express* 15, 015002 (2022).
16. Daniels, K. M. et al. Mechanism of Electrochemical Hydrogenation of Epitaxial Graphene. *J Electrochem Soc* 162, E37–E42 (2015).
17. Podlivaev, A. I. & Katin, K. P. Competition of hydrogen desorption and migration on graphene

surface in alternating electric field: Multiscale molecular dynamics and diffusion study. Appl Surf Sci 686, 162125 (2025).

Response to comments of Reviewer #2

We thank the Reviewer for taking the time to carefully read and comment our manuscript.

Response to comments from Reviewer #1

Thanks for the reply. We appreciate the efforts of the authors to answer our questions and revise their manuscript. We found a few points in the revised files that merit some attention from the authors. After that the manuscript can be published.

We thank the Reviewer for their continued support throughout the review process. Answer to the points raised below.

(1) All the electron current data were replaced in the revised manuscript with conductance values. However, the plots have similar absolute values (i.e. conductance vs current). Is this correct (Figure 1b, Figure 2, bottom panel, Figure 3c bottom panel)? Taking into example Figure 1b: the absolute value now of the conductance is the same as the current they reported in the previous version. For that to be correct, length to width ratio of the graphene device should be strictly 1:1? However, in Figure S2b, the shape of the graphene is irregular and the gold electrode is a V-shaped arrangement. Similarly in the bottom panel of Figure 2, in the revised manuscript, the red and blue curves now touches at $V_g - V_{np} = -1.2V$ but not in the original manuscript. If the authors applied a scaling factor of 1.13, what is the reason? Or are the devices not at the same scale? For the point on the blue line where the current is 441.4 nA, the corresponding conductance value is 497.6 μS .

We have a similar question concerning the lower panel of Figure 3c: the vertical axis value for the Gr-SiO₂ sample remain unchanged when switching from current to conductance, while the vertical axis values of Gr-hBN and Corrugated Gr seems scaled by a certain factor? For example, the scaling factor of the Gr-hBN curve is 0.90, while the scaling factor of the Corrugated-Gr curve is 0.50. We do not understand the origin of these scaling factors reported now in the revised manuscript.

The original draft had data in current units (nA) and the applied sd voltage is 1 mV, so the numbers look identical in μS conductance units ($1 \text{ nA}/1 \text{ mV} = 1 \mu S$). Normalising by length and width of the flakes led to the differences noted by the Reviewer.

Given that these are two-probe measurements with flakes of similar, but irregular size, our preference has always been to present the data without any normalisation—as we did in the first version. In the revised manuscript, we present all data as conductance (current/voltage) without normalisation by flake size. We believe that this is the clearest presentation.

(2) The data point of the original Figure S6 for the 10 μm pore is not shown in the new plot (Figure S6) of the new manuscript. This data point shows that the conductance of DTFSI is 0.19 μS and the conductance of HTFSI is 0.17 μS . Including these data points in the fitting of circular hole etched in the SiN_x membrane would yield a larger bulk conductance for H and D.

We calculated those 0.19 μS using the intercepts: For DTFSI curve, we read two points :(-3, -0.56), (3, 0.57). The slope is $(0.57+0.56)/(3+3)=0.19 (\mu S)$

Similarly, for HTFSI curve, we read two points : (-3, -0.49), (3, 0.51). The slope is $(0.51+0.49)/(3+3)=0.17 (\mu S)$

The devices shown in the original Fig. S6 were fabricated using 10 μm lithography masks, but overexposure during fabrication may have produced apertures larger than the nominal size. At the time, the aperture diameters were not characterised in detail, since our aim was to confirm that no large differences in conductance existed between the electrolytes. In the revised manuscript, all apertures were measured accurately, which allowed us to obtain precise and reproducible conductivity values. Because the exact sizes of the earlier devices are unknown,

they had to be excluded from the current dataset. With hindsight, we can see that these devices indeed must have had a larger aperture.

(3) Concerning the hydrogen saturation of the PdHx electrode: According to Murphy, D. W. et al, Chem. Mater., 1993, 5, 767-769, the Pd was immersed in a BH4- solution overnight while stirring. The previous study emphasized that the Pd was saturated with H; however, this article claims that Pd electrode has intermediate H loading. Why did the loading stop at intermediate state? In addition, if a Pd electrode that is already saturated with H, which means that it can only lose protons but cannot gain protons. Can it still be used as a reference electrode or counter electrode here?

Murphy *et al.* used Pd powders, which can easily approach full H loading due to their high surface area. Our experiments used a Pd foil, where hydrogen uptake is slower, leading to intermediate loading. Nevertheless, PdHx with both intermediate and saturated loading are in equilibrium with the electrolyte as shown by Flanagan and our own measurements in the previous round of comments, making them suitable as reference and counter electrodes.

(4) The calculation of the carrier mobility of Gr-BN. We proposed that the authors could calculate the carrier mobility for their three type of samples for comparison. Could the authors provide the value of Gr-BN as well?

For Gr-on-hBN devices, the mobility values are about an order of magnitude larger, in agreement with previous studies.

	Electron mobility $10^3 \text{ cm}^2 / \text{V}\cdot\text{s}$	Hole mobility $\text{cm}^2 / \text{V}\cdot\text{s}$	Contact resistance $\text{k}\Omega$	Residue carrier concentration $\text{E}^{11} \text{ cm}^{-2}$	Number of samples
Graphene-on-SiO2	3.800 ± 0.8	5.9 ± 0.7	2.2 ± 0.3	8.0 ± 1.1	8
Corrugated graphene	1.9 ± 0.5	2.8 ± 0.5	2.0 ± 0.3	6.9 ± 1.4	4
Graphene-on-hBN	20.0 ± 3.2	28.6 ± 3.8	2.2 ± 0.5	3.6 ± 0.5	4

Response to comments from Reviewer #2

We thank the Reviewer for taking the time to comment our manuscript.

Which one is the PdH_x pseudo-reference electrode?

What is this bias voltage between the two electrodes?

Does V_G refer to the bias voltage of Gate? However, the Gate in the figure is another electrode.

We thank the authors for their responses. We still have several questions after the authors have revised their manuscript.

- (1) Thanks to the authors for redrawing the circuit, which makes it clearer. The authors used a source meter (Keithley 2636B source meter, two-wire measurement) and a voltmeter (Keithley 2182A Nanovoltmeter) instead of using a potentiostat to perform CV scans. We believe that this platform, particularly the reference electrode, cannot provide feedback control for the current loop. The circuits in the references ^{1,2} seem to also use a source meter with a voltmeter. We therefore still question whether the use of a potentiostat would yield different results than using two separate Keithley instruments for determining hydrogenation overpotentials. In electrochemistry we use potentiostat as standard equipment.
- (2) In their response to point 2 the authors mention that the conductivity of deuterons in aqueous electrolytes is ~10% smaller than for protons (Arcis et). According to the CRC Handbook of Electrochemistry and Physics ³ (page 5-77), the conductivity of H⁺ is 1.4 times that of D⁺ in water at 25°C.
- (3) Separately, in Figure S6, the authors could not measure a significant difference of conductivity between H⁺ and D⁺ in HTFSI and DTFSI for SiN_x substrates with a hole without graphene, and attributed this result to device-to-device variations (see point 2, where a difference of 1.4 is reported for bulk conductivities of H⁺ and D⁺). Therefore, we question whether PdH_x and PdD_x are reliable in detecting proton or deuteron currents.
- (4) In our initial question we asked which electrochemical reactions occur on PdH_x. We asked this question because we believe that the electrochemical reactions occurring at the reference electrode are key to determine the overpotential of hydrogenation on graphene. We asked the following question: “what are the specific reaction equations that occur on PdH_x when gaining and losing electrons?” In their revised manuscript, the authors mention that “Pd + H⁺ + e⁻ -> PdH_x” is happening. What if the reaction would be “Pd + xH⁺ + xe⁻ → PdH_x. Separately, if PdH_x is saturated with H, atomic hydrogen or hydrides can no longer be inserted into the electrode. The literature report on an upper limit to the electrochemical insertion of H into Pd, which could be considered here for the discussion.⁵ Possible reactions include ⁶:

$2 H^+ + 2 e^- \xrightarrow{Pd/PdH_x} H_2$	(1)
$PdH_x + X H^+ + X e^- \rightarrow Pd + X H_2$	(2)
$PdH_x \rightarrow Pd + X H^+ + X e^-$	(3)
$H_2 \xrightarrow{Pd/PdH_x} 2 H^+ + 2 e^-$	(4)

Overall and over the time of the measurement, the composition of the reference electrode may change and therefore such reference electrode may not be used to determine the overpotential of hydrogenation on graphene. In order determine whether the reference electrode composition changes (or not), the authors could use XRD or XPS before and after the measurement.

- (5) In the lower panel of figure 3C, corrugated graphene shows a higher current compare to graphene on hBN and graphene on SiO₂; the authors carried out an error analysis and

concluded that differences in currents are attributed to device-to-device variations (now the author normalized the electronic current to the width of the graphene). We expect corrugated graphene to have lower charge carrier mobilities than less corrugated graphene. What if the authors would calculate and compare the carrier mobility for the different samples instead of the current ^{7,8}; and establish whether corrugated graphene has a better affinity for H⁺ (see also point 5).

- (6) Corrugated graphene should have much lower overpotential of hydrogenation comparing to graphene on SiO₂ and hBN. However, corrugated graphene was measured to have the similar overpotential as for graphene on SiO₂ and hBN (the potential at the peak of hydrogenation current in upper panel of Figure 3c). Similarly, corrugated graphene should have much lower overpotential at the insulating transition compared to graphene on SiO₂ and hBN. However, corrugated graphene was measured to have the maximum overpotential (the potential at the zero current transition in lower panel of Figure 3c) compare to graphene on SiO₂ and hBN. The authors explain these results as "concave areas relatively free of protons [...] leaves electronically conductive paths". However, this reason is unclear. Drawing a schematic diagram of electronically conductive paths in the concave and convex areas may be relevant. In addition, if the concave areas with conductive paths exist, why does corrugated graphene undergo the same insulating transition as graphene on SiO₂ and hBN?
- (7) The size of the washer compared to the size of graphene is very important for the fully insulation transition. So, we asked before, "Are the graphene samples studied here bigger or smaller than the opening in the SU8 washer?" The authors point out that the graphene flake must be smaller than the opening of the SU8 washer, which is important for the insulating transition. We strongly agree with this point of view. However, in a previous paper by the same research group ¹, the graphene flake seems larger than the opening in the SU8 washer, and an insulating transition is also observed. Are there here any contradictions between the two situations/articles?

References

1. Tong, J. *et al.* Control of proton transport and hydrogenation in double-gated graphene. *Nature* **630**, 619–624 (2024).
2. Cai, J. *et al.* Wien effect in interfacial water dissociation through proton-permeable graphene electrodes. *Nat Commun* **13**, 5776 (2022).
3. *CRC Handbook of Chemistry and Physics*. (CRC Press, 2014). doi:10.1201/b17118.
4. Lozada-Hidalgo, M. *et al.* Sieving hydrogen isotopes through two-dimensional crystals. *Science (1979)* **351**, 68–70 (2016).

5. Benck, J. D., Jackson, A., Young, D., Rettenwander, D. & Chiang, Y.-M. Producing High Concentrations of Hydrogen in Palladium via Electrochemical Insertion from Aqueous and Solid Electrolytes. *Chemistry of Materials* **31**, 4234–4245 (2019).
6. Moumaneix, L., Rautakorpi, A. & Kallio, T. Controlling the Reactivity and Interactions between Hydrogen and Palladium Nanoparticles via Management of the Particle Diameter. *ChemElectroChem* **10**, (2023).
7. Zhang, Z., Xu, H., Zhong, H. & Peng, L.-M. Direct extraction of carrier mobility in graphene field-effect transistor using current-voltage and capacitance-voltage measurements. *Appl Phys Lett* **101**, (2012).
8. Zhong, H., Zhang, Z., Xu, H., Qiu, C. & Peng, L.-M. Comparison of mobility extraction methods based on field-effect measurements for graphene. *AIP Adv* **5**, 057136 (2015).